# Temporal order and precision of complex stress responses in individual bacteria

Karin Mitosch[1,2], Georg Rieckh[1,3] & Tobias Bollenbach[4,*] iD

## Abstract

Sudden stress often triggers diverse, temporally structured gene expression responses in microbes, but it is largely unknown how variable in time such responses are and if genes respond in the same temporal order in every single cell. Here, we quantified timing variability of individual promoters responding to sublethal antibiotic stress using fluorescent reporters, microfluidics, and time-lapse microscopy. We identified lower and upper bounds that put definite constraints on timing variability, which varies strongly among promoters and conditions. Timing variability can be interpreted using results from statistical kinetics, which enable us to estimate the number of rate-limiting molecular steps underlying different responses. We found that just a few critical steps control some responses while others rely on dozens of steps. To probe connections between different stress responses, we then tracked the temporal order and response time correlations of promoter pairs in individual cells. Our results support that, when bacteria are exposed to the antibiotic nitrofurantoin, the ensuing oxidative stress and SOS responses are part of the same causal chain of molecular events. In contrast, under trimethoprim, the acid stress response and the SOS response are part of different chains of events running in parallel. Our approach reveals fundamental constraints on gene expression timing and provides new insights into the molecular events that underlie the timing of stress responses.

**Keywords** antibiotics; bacterial stress response; gene expression timing; microfluidics; single-cell measurements

**Subject Categories** Microbiology, Virology & Host Pathogen Interaction; Quantitative Biology & Dynamical Systems; Transcription

**Mol Syst Biol. (2019) 15: e8470**

## Introduction

Microbes live in unpredictable environments where they experience sudden environmental changes requiring unremitting adaptation.

These changes commonly trigger temporally structured gene expression responses (Gasch *et al*, 2000; Steil *et al*, 2005; Mitosch *et al*, 2017). Although some of these gene expression responses may be circumstantial or suboptimal (Price *et al*, 2013), they are generally a downstream effect of alterations in intracellular molecule concentrations that follow the change in environment. Cells have evolved specific responses to such molecular events. A case in point is the upregulation of catabolic or assimilatory genes for the replenishment of limiting nutrients following a nutrient downshift in *Escherichia coli* (Zaslaver *et al*, 2004; Gyaneshwar *et al*, 2005). Likewise, upon exposure to fresh nutrients, *Bacillus subtilis* spores awaken following a highly coordinated response program that reflects physiological needs: Proteins required for gene expression are activated early, followed by biosynthesis of metabolic and cell division proteins (Sinai *et al*, 2015). We recently showed that sudden sublethal antibiotic stress triggers diverse, temporally structured gene expression changes in *E. coli* when measured at the population level (Mitosch *et al*, 2017). For instance, the prodrug nitrofurantoin (NIT), a first-choice drug against uncomplicated urinary tract infections (McQuiston Haslund *et al*, 2013), leads to the formation of nitro anion radicals and rapidly induces an oxidative stress response, followed by an SOS response to DNA damage (Bryant & McCalla, 1980) after several hours (Fig 1A). The folate biosynthesis inhibitor trimethoprim (TMP) induces several stress responses including an early acid stress response due to adenine depletion and a later SOS response (Mitosch *et al*, 2017). Most investigations of stress response dynamics have focused on the population level, averaging responses over many cells (Gasch *et al*, 2000; Steil *et al*, 2005; Dudin *et al*, 2013).

It is largely unknown if the temporal order of gene activation observed at the population level correctly reflects the temporal order in every single cell (Fig 1B). Alternatively, the temporal order measured at the population level may hold for the majority of cells, but could still be reversed in individual cells (Fig 1C); for example, this can happen when timing variability is high and uncorrelated between the two genes. Even for most individual genes, the variability of their response timing under stress and the biophysical constraints that determine this variability is unknown. Addressing these issues is crucial for interpreting the observed timing variability and for elucidating the temporal sequence of molecular events that occur inside cells. Importantly, any interpretation that the temporal

1 IST Austria, Klosterneuburg, Austria
2 EMBL Heidelberg, Heidelberg, Germany
3 Division of Biological Sciences, University of California at San Diego, La Jolla, CA, USA
4 University of Cologne, Cologne, Germany
 *Corresponding author. Tel: +49 221 470 1621; E-mail: t.bollenbach@uni-koeln.de

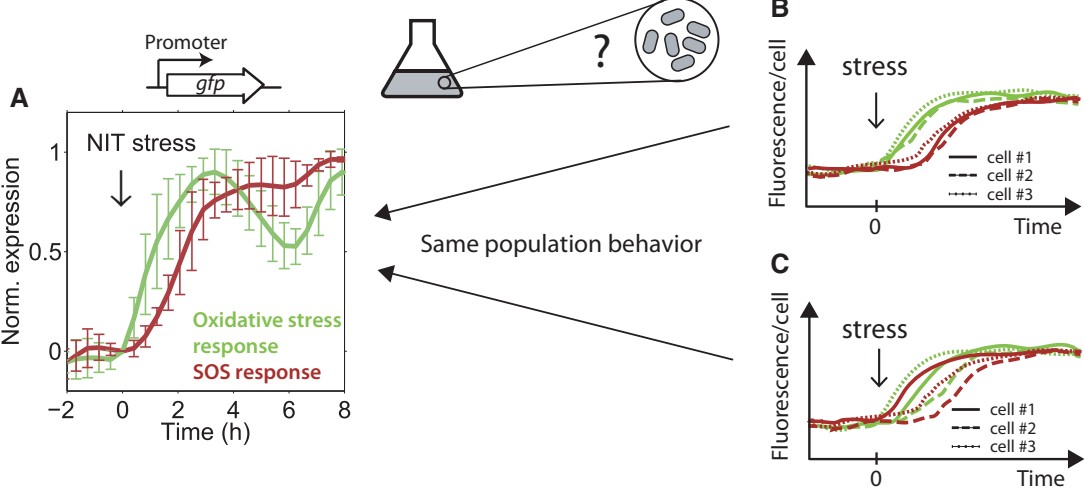

**Figure 1. The temporal order of stress responses observed at the population level does not necessarily reflect the temporal order in every single cell.**

A   Green line: Normalized population expression level averaged over all OxyR, and SoxS-regulated promoters; red line: average over all LexA-regulated promoters (Mitosch *et al*, 2017), as measured with a promoter-GFP plasmid library (Zaslaver *et al*, 2006) in a plate reader. Oxidative stress promoters clearly precede SOS response promoters in response to NIT stress, when measured at the population level (Mitosch *et al*, 2017). Lines show the mean and error bars show the standard deviation over seven oxidative stress and SOS promoters, respectively (Materials and Methods). It is not clear if this temporal order correctly reflects the temporal order in single cells.

B   Schematic showing response of two different genes (green and red) in three different cells (solid, dashed, and dotted line). The temporal order observed at the population level correctly reflects the temporal order in each individual cell.

C   As (B) but here, the temporal order at the population level is not the same in every single cell: Although in most cells, the green gene responds before the red one, one cell expresses the red gene before the green one (solid lines).

order of the responding genes evolved for a particular function (Zaslaver *et al*, 2004) requires that the response of individual cells actually obeys this temporal order. To reveal if that is the case, dynamic measurements of multiple genes in the same cell are needed. Such measurements were previously used to analyze dynamic gene expression correlations in isolated genetic circuits (Locke & Elowitz, 2009), but their potential for probing connections between genes in complex stress responses remains unexplored.

Here, we systematically investigated timing variability of individual promoters and the temporal order of gene expression in response to antibiotic stress in single *E. coli* cells. We found that timing variability generally increases with the response time and is constrained by both a lower and an upper bound, which we interpret using statistical kinetics. These bounds increase linearly: Every increase in mean response time by 1 h increases the timing variability by at least 10 min. The SOS response to DNA damage is particularly interesting: It shows low response time variability under NIT but is more variable under TMP stress. To elucidate if the SOS response is causally linked to earlier stress responses, as suggested by the clear sequential order observed in population-level measurements, we developed a method for the rapid and efficient construction of dual-reporter strains that enable the simultaneous readout of two responses in the same cell. We found that each individual cell under NIT stress first triggers the oxidative stress response, followed by the SOS response in a strikingly clear temporal order with highly correlated response times. The oxidative stress and the SOS response are therefore likely part of the same chain of molecular events triggered in response to NIT. In contrast, such a temporal order and response time correlation are absent for the acid stress and SOS responses under TMP stress, suggesting that these two stress responses are constituents of independent chains of molecular events. Overall, we show that measuring the response dynamics of multiple genes in individual cells is a powerful underutilized approach for testing specific hypotheses for the preceding sequences of molecular events that ultimately activate the stress responses.

## Results

### Every increase in mean response time by 1 h increases timing variability by at least 10 min

To systematically address how precise the timing of stress responses is at the single-cell level, we first quantified timing variability for individual promoters in different antibiotic stress conditions (Fig 2). We measured the expression of 23 different chromosomally integrated promoter–fluorescent protein (FP) constructs in single cells inside a microfluidics device using time-lapse microscopy (Materials and Methods; Table EV1). Based on our previous population-level measurements (Mitosch *et al*, 2017), we selected promoters from a genome-wide GFP reporter library (Zaslaver *et al*, 2006) that were strongly induced in response to at least one of the antibiotics TMP, NIT, and tetracycline (TET): TMP inhibits folate biosynthesis (Hitchings & Smith, 1980), NIT is a prodrug that is reduced and thereby activated by intracellular nitroreductases to reactive compounds which damage macromolecules (Bryant & McCalla, 1980), and TET inhibits the small subunit of the ribosome (Pioletti *et al*, 2001). All three drugs provoke massive genome-wide gene expression changes (Mitosch *et al*, 2017). Chromosomal

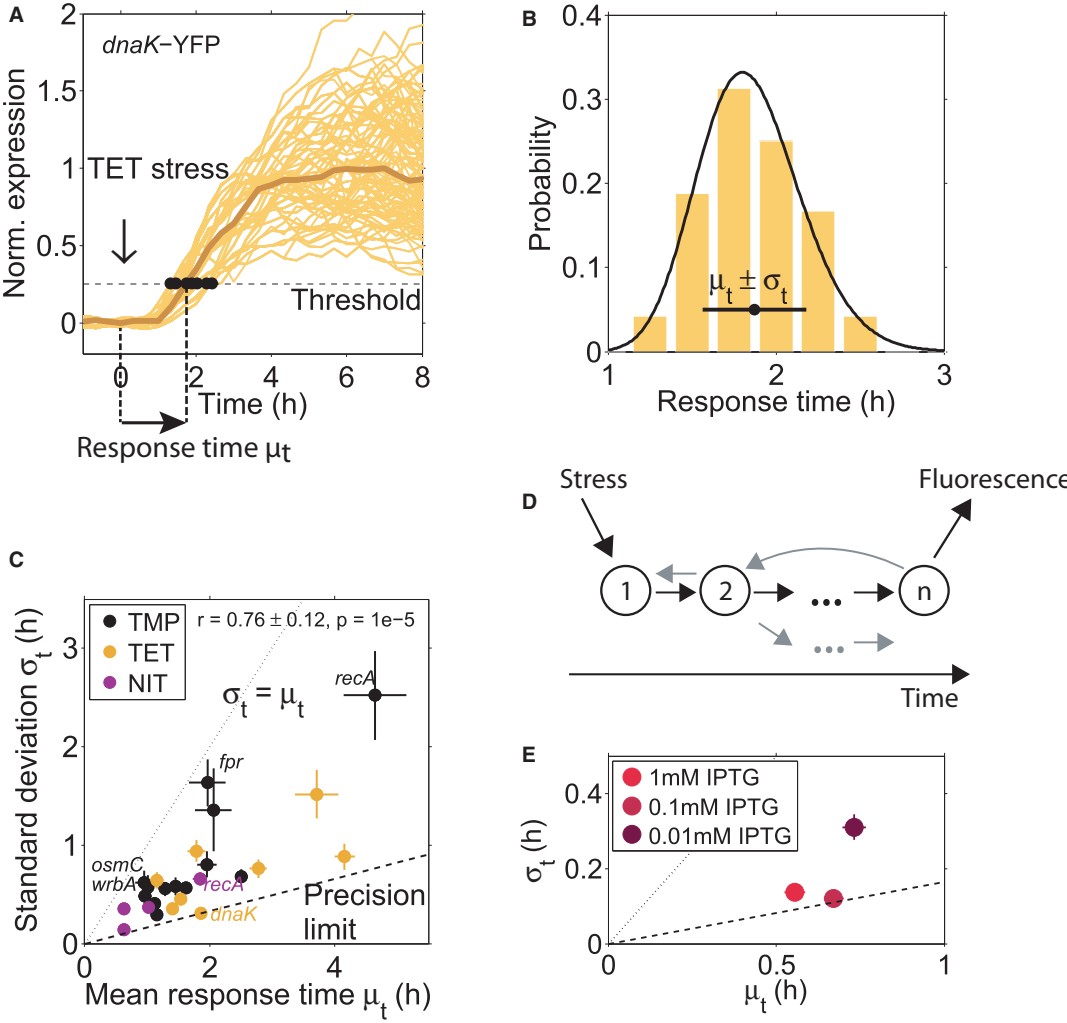

**Figure 2. Hard limits constrain the timing variability of individual promoters from above and below.**

A   Single-cell gene expression time traces from the *dnaK* promoter in response to TET stress, normalized to the median full response. Response times were determined as the time point at which a threshold expression level was reached (see main text and Materials and Methods). This threshold was low enough so that most cells exceed it and high enough to avoid false positives due to the low signal-to-noise ratio at time point zero. Brown line: median of all cells. Time traces are from one microcolony.

B   Histogram of the response times for the *dnaK* promoter with mean $\mu_t = 1.87$ h and standard deviation $\sigma_t = 0.37$ h, and fit of an Erlang distribution with shape parameter $n = 37$ (black line; see text and Materials and Methods). Response times are from two microcolonies.

C   Standard deviation $\sigma_t$ versus mean response time $\mu_t$ for 23 different promoters (Table EV1) in three antibiotic stress conditions (TMP, TET, and NIT). The standard deviation of the response time $\sigma_t$ grows with the mean response time $\mu_t$ and does not fall below a "precision limit" (dashed line) that increases linearly with a slope ~ 0.165. The dotted line indicates the upper bound to timing variability where $\sigma_t = \mu_t$, see text. The promoter *dnaK* under TET has low timing variability, whereas the promoters *recA*, *fpr*, *osmC*, and *wrbA* under TMP stress have high timing variability. The response time mean and standard deviation are from subsampling of descendants of single cells that were present at the time of stress addition (Materials and Methods). Subsampling for each promoter was done from at least two microcolonies, and the descendants of at least 17 individual cells present at the time of stress addition.

D   Schematic of a molecular chain of *n* events triggered by stress and resulting in measurable fluorescence, according to the statistical kinetics model applied here. The gray arrows indicate possible deviations from a linear chain of events, like feedback, reversibility, or branching, which are also captured in the model.

E   Standard deviation $\sigma_t$ versus mean response time $\mu_t$ for the *LlacO-1* promoter induced with different IPTG concentrations. Note that all cells crossed the defined threshold under the tested IPTG concentrations. The dashed and dotted lines are the same as in (C).

integration is crucial for studies at the single-cell level since fluctuations or systematic changes in plasmid copy number otherwise complicate the measurements (Dunlop *et al*, 2008; Bollenbach & Kishony, 2011). For chromosomal integration, we selected promoters with a sufficiently high expression level to ensure the reliable detection of dynamic expression changes.

These constructs enabled precise measurements of response times in single cells. After the cells in the microcolony underwent several doublings, drugs were suddenly added (time zero) at concentrations that result in ~ 50% growth rate inhibition. The average growth rate was relatively stable after drug addition (Fig EV2A), but varied from cell to cell. For each cell and promoter, we determined

the response time as the time point at which expression exceeded a threshold of 25% of the median fully induced expression in the microcolony (Materials and Methods, Fig 2A and B). For each promoter, we combined data from at least two microcolonies measured on the same day. To quantify the typical time that passes before a promoter responds to the stressor and the cell-to-cell variability of this time, we determined the mean and the standard deviation of the response times from individual cells, respectively (Fig 2B).

The temporal precision of stress responses in individual cells has a clear limit that depends on how much time has passed since the addition of the stressor. We first noticed that the standard deviation of the response times correlated strongly with the mean response time (Fig 2C; $r = 0.76 \pm 0.12$, $P = 1 \times 10^{-5}$). Thus, there was a clear trend that the timing of later responding promoters was more variable in absolute terms (Fig 2C), consistent with previous single-cell studies (Amir *et al*, 2007; Megerle *et al*, 2008). This trend is similar for all three stressors TMP, NIT, and TET (Fig 2C) and is not a simple consequence of the cell-to-cell variability in growth rate (Fig EV2A, B and E). It is plausible that promoters that respond later exhibit greater absolute timing variability: Errors in the timing of transcription factor activation or metabolic reactions from preceding processes may accumulate, as has been shown for the cell-to-cell variability of expression levels (Pedraza & van Oudenaarden, 2005). Our analysis further suggested a fundamental limit for the precision at which cells can control the timing of stress responses: Many promoters in Fig 2C approach a line with a slope of ~ 0.165 ("precision limit"), but no promoter beats this precision limit. Determining the response time with an alternative measure, specifically the time until 50% of maximum expression was reached for each individual cell, did not appreciably alter these observations (Fig EV2C and D). Thus, for every hour that passes after stress addition, the standard deviation of the response time of any response triggered by the cell increases by at least 10 min.

**Precision limit and number of underlying rate-limiting steps can be estimated using statistical kinetics**

To interpret the observed dependence of timing precision on response time, we use a general model from statistical kinetics. In this model, the addition of a stressor at time zero triggers a sequence of discrete molecular steps that occur inside the cell; these steps may be reversible, and the sequence can include branches and feedback loops (Fig 2D). There can be many such steps, and in general, we do not know these molecular steps. We can think of them as events such as the uptake of the stressor molecule into the cell, the binding of a transcription factor to its binding site on the DNA, or a sudden drop in intracellular pH, but also the production of multiple copies of the same mRNA or protein may be viewed as several steps. The completion of this chain of molecular steps ultimately leads to the observed gene expression response. Due to inevitable molecular noise, each step is stochastic and takes a variable time. Assuming that each molecular step is memory-less with an exponential waiting time distribution (Materials and Methods, Fig EV1), a central result of statistical kinetics is that the standard deviation of the response times $\sigma_t$ divided by the mean response time $\mu_t$ is equal to or greater than the inverse of the square root of

the number of rate-limiting steps $n$ (Aldous & Shepp, 1987; Moffitt & Bustamante, 2014).

$$\frac{\sigma_t}{\mu_t} \geq \frac{1}{\sqrt{n}} \qquad (1)$$

This inequality becomes an equality when the process is a strictly linear sequence of steps that are irreversible and happen at exactly the same rate; however, inequality (1) holds more generally, including for reversible reactions, branches, or loops. Importantly, while $n$ is a very conservative lower bound for the number of steps, it can be interpreted as an estimate for the effective number of rate-limiting steps (Moffitt & Bustamante, 2014), i.e., the number of relatively slow molecular reaction steps that lie between the addition of the stressor and the detection of the fluorescent reporter signal for a gene expression response. All these steps need to be similarly slow —if one step becomes much slower than the others, that step alone becomes rate-limiting, which leads to increased timing variability. Processes with a greater number of rate-limiting steps $n$ can achieve more precise timing (for more details, see Materials and Methods; Fig EV1). Note that we cannot detect single molecules in our experiments. Therefore, several mRNA and protein molecules have to be produced until the threshold (25% of the median fully induced expression level) is crossed. This can correspond to several steps in the model. The previously unexplained linear increase in the precision limit with mean response time (dashed line in Fig 2C) follows immediately from inequality (1) if the number of rate-limiting steps $n$ is fixed.

We corroborated that our data are consistent with this model by validating that the measured response times followed an Erlang distribution (Fig 2B, Materials and Methods), which is the distribution of the completion times predicted by the model (Materials and Methods). Assuming a linear chain of rate-limiting steps, our data therefore suggest that there is a general lower bound, the "precision limit," for the number of rate-limiting steps underlying stress responses in *E. coli*, which holds across diverse promoters and stressors (Fig 2C). From the slope of the dashed line in Fig 2C, which corresponds to $\sigma_t/\mu_t \approx 0.165 \pm 0.012$, we estimate that the most precisely timed responses require underlying molecular reactions with $n \geq 37 \pm 9$ rate-limiting steps. A precisely timed response could result from a high number of proteins per cell due to a high promoter strength (Co *et al*, 2017). If this were the predominant reason for high precision among the promoters we investigated, there should be a clear negative correlation between absolute protein expression level and timing variability. However, no such correlation is apparent in our data (Fig EV2H, $r = -0.08 \pm 0.23$).

Note that the number of rate-limiting steps estimated using the model from statistical kinetics cannot be determined with high precision from our data. In principle, cells could achieve higher timing precision by using even more rate-limiting steps, but there is no indication for this in our data set. These ~ 37 rate-limiting steps must have approximately equal duration, as a large deviation in a single step would render that step alone rate-limiting. On average, a rate-limiting step takes 1–3 min for the earliest and most precise promoters; all rate-limiting steps for the promoters we observed therefore take at least that long and for many promoters, the rate-limiting steps are considerably slower. Based on this estimate, we

conclude that elongation steps in transcription or translation of the fluorescent reporter protein cannot be rate-liming, as the addition of single nucleotides or amino acids takes on the order of tens of milliseconds (Bremer & Yuan, 1968; Young & Bremer, 1976). Transcription initiation, however, is a much slower process (on a time scale of minutes) and may therefore be a rate-limiting step for many observed promoters (McClure, 1980; Hammar *et al*, 2012). Since we cannot detect the first fluorescent protein produced, the production of several copies of the same molecule (e.g., the fluorescent reporter protein) may also contribute multiple rate-limiting steps. Although this reduces the informative value of the estimated number of rate-limiting steps, this quantity can still indicate simpler responses that have few such steps.

We hypothesized that interfering with promoter activation could change the number of rate-limiting steps. While this is challenging for the complex stress responses to antibiotics, we can directly control the kinetics of the activation of the IPTG-inducible *LlacO-1* promoter by varying the inducer concentration. At high inducer concentrations, timing variability of this promoter was close to the limit of precision (Fig 2E). When inducing this promoter with very low IPTG concentrations (0.01 mM), the mean response time increased. Importantly, the standard deviation of the response time increased disproportionally (Fig 2E). This suggests that relatively few steps in the chain of events, such as the uptake of IPTG into cells, become sufficiently slow to turn into the only effectively rate-limiting steps at such low inducer concentrations. Overall, this experiment can be understood using the statistical kinetics framework and illustrates how increased average response times can coincide with fewer rate-limiting steps.

Importantly, since there must be at least one rate-limiting step, inequality (1) also sets an upper bound: The standard deviation $\sigma_t$ cannot exceed the average response time $\mu_t$ (dotted line in Fig 2C). Consistent with this prediction, some promoters in our data set (e.g., *recA*, *fpr*, *osmC*, *wrbA* under TMP stress) approach this upper bound from below, but no promoter in any condition exceeds it (Fig 2C). Since only a few rate-limiting steps determine the response of such promoters, the identification of these steps with targeted experiments may be feasible. Taken together, this analysis establishes a quantitative foundation and a baseline for the comparison of timing variability between promoters with different mean response times; it further shows how observing response timing in single cells can be used to estimate the number of rate-limiting steps that underlie these responses.

A specific result from this analysis is that relatively few molecular steps seem to determine the timing of the SOS response promoter *recA*. In addition, the *recA* promoter showed differing behavior in two stress conditions: Its response was relatively precisely timed under NIT ($\sigma_t/\mu_t = 0.36$) but had higher timing variability under TMP ($\sigma_t/\mu_t = 0.54$; Fig 2C). Applying equation (1) and rounding up, we estimate eight effective rate-limiting molecular steps for NIT stress and only four steps for TMP stress. The fact that there are fewer rate-limiting steps under TMP stress suggests that the sequence of molecular events leading to the activation of *recA* is markedly different in both conditions. As LexA is the only known transcriptional regulator of *recA* (Keseler *et al*, 2017), we hypothesized that the difference in *recA* activation between the two stressful conditions stems from events further upstream such as the occurrence of DNA damage.

## Cloning method enables efficient chromosomal integration of promoter pairs

The sizable variability in response timing observed for *recA* raises the question if these gene expression changes are precisely timed with respect to each other in individual cells: In principle, each cell could go through a strictly ordered sequence of molecular events and corresponding responses, even if the timing overall varies strongly from cell to cell (Fig 1C). To probe potential upstream molecular events and their temporal order with respect to the SOS response, we combined reporters for pairs of promoters in the same cell. We enhanced the method for the chromosomal integration of single promoters from the promoter-GFP library (Zaslaver *et al*, 2006) for the efficient integration of promoter pairs (Fig 3). Our method uses lambda-red recombineering (Datsenko & Wanner, 2000) for the initial chromosomal integration of "platforms" (Fig 3A–C; Materials and Methods). These platforms then provide the basis for the integration of promoters from the promoter-GFP plasmid library (Zaslaver *et al*, 2006). We validated that both YFP and CFP reliably reported on the timing variability of promoters by integrating the IPTG-inducible promoter *LlacO-1* upstream of both YFP and CFP. Response times obtained with YFP and CFP were highly correlated and had similar averages ($r = 0.81 \pm 0.04$, $P = 6 \times 10^{-12}$; Fig 3D). Similar results were obtained for another promoter (*gadB; $r = 0.88 \pm 0.05$, $P = 7 \times 10^{-6}$*; Fig EV3A) and for swapped fluorescent proteins ($r = 0.64 \pm 0.06$, $P = 9 \times 10^{-9}$; Fig EV3B). CFP has a lower signal-to-noise ratio due to cellular autofluorescence in the blue channel (Monici, 2005; Mihalcescu *et al*, 2015). Consequently, we used CFP only for sufficiently strong promoters. Overall, these data show that both YFP and CFP can be used to precisely determine response times from single cells and that their combination in the same cells can report on temporal shifts and correlations between different promoters.

## The oxidative stress response strictly precedes the SOS response in every single cell under NIT stress

We investigated temporal order under NIT stress at the single-cell level using dual-reporter strains constructed in this way. When measured at the population level, reporters for oxidative stress (*ybjC*) and SOS response (*recA*) showed that the oxidative stress response clearly preceded the SOS response by about 1 h (Fig 4A). At the single-cell level, response time measurements as in Fig 2 confirmed that, on average, oxidative stress precedes DNA stress. However, this timing varied considerably from cell to cell (Fig 4B). As a result, the two response time distributions for *ybjC* and *recA* overlap and we cannot determine whether the temporal order of the responses is conserved in every single cell based on the variability of both promoters alone. The dual-reporter strain with both promoters in the same cell revealed that the oxidative stress response was activated strictly before the SOS response in every single cell (Fig 4B and C). Such a strict temporal order is unlikely to occur due to random chance (permutation test, $P = 0.017$). This strict temporal order is remarkable since no connection from the oxidative stress to the SOS response is known at the level of transcriptional regulation (Keseler *et al*, 2017).

Further, response times for oxidative stress and SOS promoters were strongly correlated ($r = 0.74 \pm 0.04$, $P = 3.8 \times 10^{-6}$), i.e., an

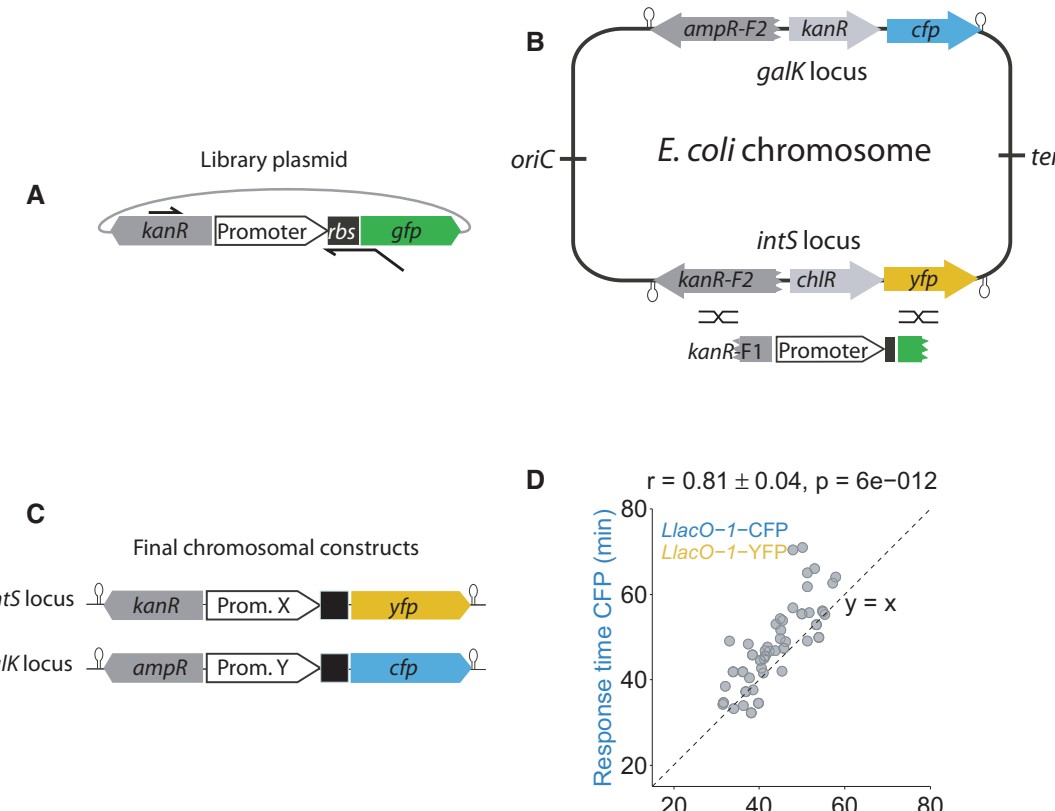

**Figure 3.   Dual-reporter method enables precise determination of response time correlations for promoter pairs.**

A   Schematic showing the primers used to make the PCR product from the library plasmid (Zaslaver *et al*, 2006).

B   Schematic of "platforms" integrated into the *intS* and the *galK* locus of the *Escherichia coli* chromosome. These platforms consist of the sequence for a fluorescent protein (*yfp* or *cfp*), a functioning resistance marker (*chlR* or *kanR*), and a truncated and thus defunct resistance marker (*kanR* or *ampR*). A PCR product shown in (A) was introduced by recombineering (black crossed lines) into the platforms.

C   Schematic of final reporter constructs in the chromosome. The functional resistance marker is replaced by the promoter of interest, and the defunct resistance marker is completed, yielding promoters driving *yfp* and *cfp*, respectively.

D   Response times of the *LlacO-1* promoter (Lutz & Bujard, 1997), induced with 1 mM IPTG at time zero, measured with YFP and CFP constructs in the same cell ($r = 0.81 \pm 0.04$, $P = 6 \times 10^{-12}$).

early *ybjC* response generally coincided with an early *recA* response. This strong correlation suggests that the detection of oxidative stress is one of the preceding molecular events before the SOS response under NIT. We observed this correlation and the sequential temporal order consistently in different microcolonies and in three replicates on different days (Fig EV4A and Table EV2). Another oxidative stress reporter (*fpr*) also showed the clear temporal order with *recA* (Fig EV5C). Together, these results show that the temporal order at the population level indeed reflects a clear order at the single-cell level for these promoters under NIT stress. Bacteria therefore invariably sense oxidative stress before DNA damage, suggesting that this oxidative stress is a key causal event in the molecular sequence leading to DNA damage under NIT.

As this result is solely based on correlations, we aimed to substantiate the causal role of oxidative stress in downstream cell damage under NIT using an independent approach. To this end, we tested the growth of oxidative stress mutants in the presence of NIT. Three out of six tested deletion mutants (Δ*sodA*, Δ*gshA*, and Δ*gshB*) showed a clear and specific growth defect under NIT stress which could be rescued by complementation (Fig EV5D and E). As these mutants have a defect in counteracting oxidative stress, the observed growth defect suggests that oxidative stress plays a crucial role for downstream cell damage under NIT. We further measured the expression of *ybjC* and *recA* in the *gshA* knockout mutant, which has impaired ability to counteract oxidative stress, in response to NIT. This mutant exhibited an amplified oxidative stress response, followed by a stronger DNA stress response (Fig EV5F and G), providing additional evidence for a causal role of oxidative stress in DNA stress under NIT.

**The acid stress response to TMP shows no strict temporal order with respect to the SOS response in single cells**

Motivated by this encouraging result for NIT, we next aimed to probe temporal order with the SOS response under TMP stress and thereby identify molecular events upstream of DNA damage that control the timing of the SOS response under TMP. Due to the high timing variability of *recA* under TMP stress, a clear

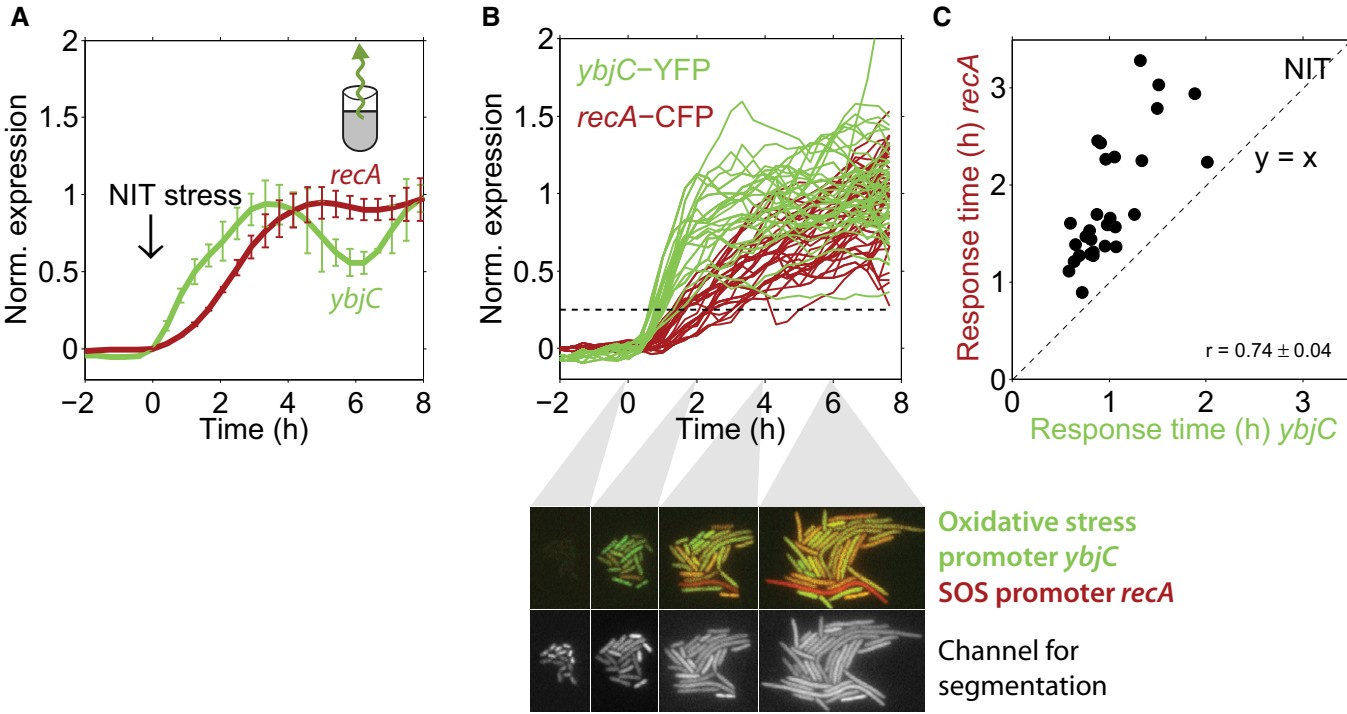

**Figure 4. Early oxidative stress response under NIT is precisely timed and precedes the SOS response in every single cell.**

A Normalized expression from the promoters *ybjC* and *recA* in response to NIT stress, measured at the population level using GFP reporters (Zaslaver *et al*, 2006); 4 μg/ml NIT was added at time zero. Data are taken from Mitosch *et al* (2017). The oxidative stress promoter *ybjC* clearly precedes the SOS response promoter *recA*, when measured at the population level (Mitosch *et al*, 2017). Error bars show standard deviation of three replicate experiments measured on different days.

B Normalized *ybjC* and *recA* expression over time in response to the addition of 4 μg/ml NIT at time zero in single cells from one microcolony. Dashed line: threshold used to determine response times (Materials and Methods). Lower panels, upper row: dual-color images of the oxidative stress reporter *ybjC* (green) and the SOS reporter *recA* (red) in single cells at four different time points after NIT addition in one microcolony. Note that the *ybjC* signal (green) is stronger at *t* = 2 h, whereas the *recA* signal (red) is only apparent at *t* = 4 and 6 h. Lower row: Constitutively expressed mCherry used for segmentation.

C Response times for *ybjC* and *recA* from individual cells with dual reporters (Fig 3), combined from two microcolonies. All response times for *recA* are higher than for *ybjC*, and response times for both reporters are strongly correlated ($r = 0.74 \pm 0.04$, $P = 3.8 \times 10^{-6}$). The error of the correlation coefficient is from subsampling of descendants of single cells that were present at the time of NIT addition (Materials and Methods).

temporal order with *recA* at the single-cell level is only attainable if a promoter responds very early or exhibits response time variability that is correlated with *recA* (Fig 1). The highly variable timing of *recA* (Fig 2C) suggested that there are only about four rate-limiting molecular events leading to SOS induction. As a first candidate for one of these relevant upstream events, we focused on the acid stress response, which is rapidly and strongly induced in response to TMP (Mitosch *et al*, 2017), about 3 h before the SOS response when measured at the population level (Fig 5A). However, using a dual-reporter strain for acid stress (*gadB*) and SOS response (*recA*), we found that this order was not conserved in every single cell: A sizable fraction of cells (four out of 31) activated *recA* before *gadB* (Fig 5B and C) and response times of both promoters did not correlate ($r = 0.08 \pm 0.08$; $P = 0.68$; Fig 5C). The specific promoters used for reporting on acid and DNA stress, respectively, may introduce some of the timing variability. Still, these data provide evidence that acid and DNA stress in response to TMP belong to different and largely independent causal chains of molecular events.

We corroborated this point using artificial growth conditions that suppress the acid stress response under TMP. If the SOS response still occurs even if there is no prior acid stress that would virtually rule out any causal role of the latter in the former. TMP inhibits FolA (dihydrofolate reductase; DHFR), a key enzyme in folate biosynthesis, which leads to the depletion of purine bases and thymidine (Miovic & Pizer, 1971). We recently showed that the depletion of purines alone causes the activation of the acid stress response (Mitosch *et al*, 2017). Thus, we suppressed the acid stress response to TMP by supplementing the growth medium with the purine inosine. Appropriately, cells did not induce the acid stress response (*gadB*) anymore, but the SOS response (*recA*) was still strongly induced (Fig 5D). This experiment demonstrates that the acid stress and SOS response belong to independent chains of molecular events under TMP stress (Fig 6), explaining the clear lack of temporal order and response time correlations between these responses (Fig 5C).

# Discussion

A fundamental goal of biology is to reveal the key molecular events and their relationships that govern cellular adaptations. This

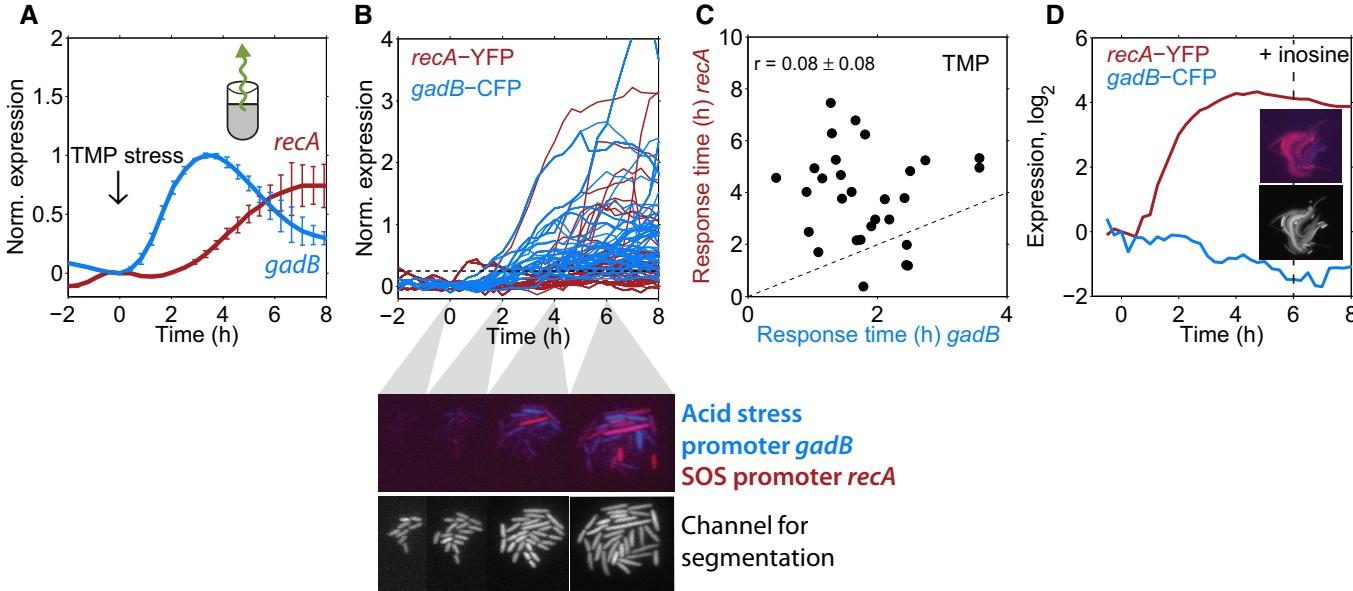

**Figure 5. Early acid stress under TMP is not part of the causal chain of molecular events leading to DNA damage.**

A Expression from the *gadB* and *recA* promoters, measured at the population level using GFP reporters (Zaslaver *et al*, 2006) over time; 0.5 μg/ml TMP was added at time zero. The acid stress response (*gadB*) clearly precedes the SOS response (*recA*). Error bars show standard deviation of three replicate experiments measured on different days.

B Normalized expression of *gadB* and *recA* expression over time in individual cells responding to the addition of 0.5 μg/ml TMP at time zero shown for one microcolony. Dashed line: threshold used to determine response times (Materials and Methods). Lower panels, upper row: dual-color images of *gadB* (blue) and *recA* (red) in single cells at four different time points. Most cells are predominantly either red or blue, i.e., acid stress and SOS response are typically not both strongly expressed in the same cells. Lower row: Constitutively expressed mCherry used for segmentation.

C Scatter plot of response times for *gadB* and *recA* in single cells; there is no clear temporal order and no significant correlation ($r = 0.08 \pm 0.08$, $P = 0.68$); data are combined from three microcolonies. Error is from subsampling of descendants of single cells present at the time point of TMP addition (Materials and Methods).

D Expression of *gadB* and *recA* in inosine-supplemented medium, averaged over 13 single cells in a microcolony. Insets: Dual-color image of *gadB* and *recA* and segmentation image as in B at $t = 6$ h.

goal is commonly tackled using genetic perturbations (Jarvik & Botstein, 1973; Bar-Joseph, 2004). We established a dual-reporter method that can disentangle molecular sequences of events and hint at causal connections between processes during cellular adaptations. Importantly, this method can suggest causal connections not only at the transcriptional network level, but also on higher-level connections between molecular processes. The technique is minimally invasive as it uses naturally occurring fluctuations instead of genetic or other perturbations to reconstruct molecular events (Dunlop *et al*, 2008; Munsky *et al*, 2009; Wong *et al*, 2011; Stewart-Ornstein *et al*, 2012). Our application of this method to bacterial stress responses to antibiotics gave unprecedented insights into the organization of these complex stress responses; by identifying the contributing molecules and their interactions, we may be able to modify bacterial responses to antibiotics and prevent undesired treatment outcomes. It is, however, important to note that a clear sequential order in response to stress does not immediately imply causality of molecular events; for example, two promoters might be activated sequentially because the (unrelated) causes for their activation are sufficiently far apart in time. In these cases, the presence or absence of response time correlations provides additional support for or against a causal relation. A limitation to keep in mind is that high intrinsic noise (Elowitz *et al*, 2002) in the expression of the reporters used might artificially increase timing variability and thus mask any temporal order or response time correlations.

Our estimates suggest that the fastest rate-limiting steps happen on a time scale of minutes, indicating that the addition of single nucleotides or amino acids in transcription or translation is not rate-limiting. Transcription initiation (McClure, 1980), the transition from initiation to elongation (Reppas *et al*, 2006), or the maturation of fluorescent proteins (Balleza *et al*, 2017), however, may well be rate-limiting steps and contribute to the high number of rate-limiting steps estimated for some promoters. For promoters approaching the upper bound of timing variability, expression is controlled by few rate-limiting steps, and in some cases, it approaches the limit of a single step (Fig 2C). Although expression from these promoters depends on many more molecular steps, their timing variability is governed by just a few rate-limiting events. In these cases, identifying the effective rate-limiting steps and thus elucidating the underlying molecular chain of events appear feasible. In general, key rate-limiting steps could include the entry of the antibiotic into the cell and the initiation of transcription. For TMP, the acidification of the cytosol could occur on a timescale of minutes (Mitosch *et al*, 2017) and thus be a rate-limiting step in the chain of events triggering the acid stress response. For the *recA* promoter under TMP stress (Fig 5), we speculate that nucleotide depletion, leading to stalling of

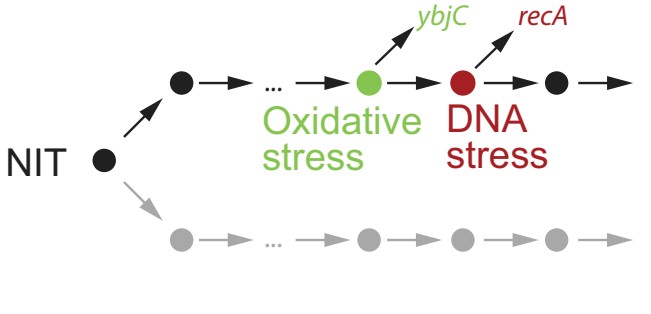

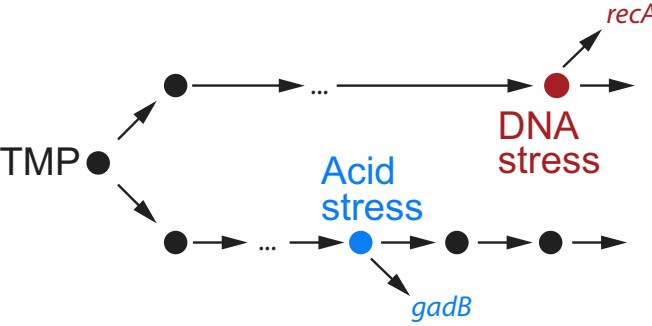

**Figure 6.  Schematic of the different chains of molecular events leading to DNA stress under NIT and TMP.**

Big dots indicate rate-limiting molecular events; arrows indicate connections between these events. Both antibiotics, NIT and TMP, induce DNA stress, reported here by the *recA* promoter. For NIT, oxidative stress (reported here by the *ybjC* promoter) is likely upstream in the same chain of molecular events as DNA stress. Other chains of molecular events (gray) that may run in parallel were not probed here. For TMP, DNA stress is preceded by fewer rate-limiting events compared to NIT, and it is not in the same chain as acid stress (reported here by the *gadB* promoter); both are generated by different upstream molecular events that were not probed here.

replication forks, is one of the rate-limiting events toward DNA damage. The formation of RecA filaments on single-stranded DNA at the forks, which is needed for the activation of the SOS response, may be another rate-limiting step—it could be slow due to the low number of replication forks where DNA damage occurs under nucleotide limitation (Giroux *et al*, 2017).

Under NIT stress, nitro anion radicals, produced by reduction of the prodrug NIT to its active form, likely activate oxidative stress transcription factors (Pomposiello & Demple, 2001). Under aerobic conditions, the nitro anion radical quickly transfers its electron to oxygen resulting in superoxide and subsequent oxidative stress. If these reactive species cannot be removed by the cell, they cause DNA damage including cross-links (Sengupta *et al*, 1990) which activates the SOS response (Imlay, 2013). There is no known regulatory connection at the transcriptional level from the oxidative stress to the SOS response (Keseler *et al*, 2017). According to the central result from statistical kinetics, inequality (1), a reporter that directly reports on a preceding rate-limiting event should have fewer rate-limiting steps and therefore a higher ratio of $\sigma_t/\mu_t$ than a reporter for a successive event. Using single-cell data, such a relation was indeed observed between the promoter controlling expression of lytic genes and bacterial lysis in the lytic cascade of bacteriophage λ (Amir *et al*,

2007). In our data, we found a slightly higher $\sigma_t/\mu_t$ ratio for *ybjC* compared to *recA* in 1/3 replicate experiments and ratios for both promoters correlated (Fig EV4D). These data alone do not imply that fewer rate-limiting steps precede expression from the *ybjC* promoter than expression from the *recA* promoter. A likely explanation for this observation is rather that, although *ybjC* is activated by a further upstream event (oxidative stress) compared to *recA*, its activation still involves several steps (activation of its transcription factor(s) by oxidative stress, transcription factor binding, transcription initiation, etc.) before the reporter signal is detected; these steps may be rate-limiting and can therefore decrease variability. The strong positive correlation between *ybjC* and *recA* response times strongly suggests that oxidative stress precedes DNA damage in a causal chain of molecular events (Fig 6).

Our analysis of response time correlations at the single-cell level is broadly applicable to diverse other organisms and conditions. It can distinguish simpler responses that depend on only a few rate-limiting steps from more complicated responses that require many such steps. Overall, the approach presented here can thus help to focus research efforts on responses where a detailed elucidation of all key steps that control the timing of the response is feasible. It thus establishes a quantitative foundation for gaining deeper insights into the dynamics and causality of key molecular events during cellular adaptation.

# Materials and Methods

### Bacterial strains, antibiotics, and culture conditions

We used the *E. coli* K-12 strain MG1655 as wild type, unless stated otherwise. This strain has been found to have a large deletion around the global transcription factor *fnr* which has been described before (Soupene *et al*, 2003). The six deletion strains for an initial growth defect check on NIT (Δ*grxA*, Δ*gshA*, Δ*gshB*, Δ*sodA*, Δ*fpr*, and Δ*oxyR*) were taken from the Keio library (Baba *et al*, 2006). The deletion strains Δ*nfsA*, Δ*sodA*, Δ*gshA*, and Δ*gshB* were P1-transduced from the KEIO collection (Baba *et al*, 2006) into the MG1655 background and PCR verified using a primer binding inside the kanamycin resistance CAGTCATAGCCGAATAGCCT (Datsenko & Wanner, 2000) and primers binding upstream of the respective open reading frames: Δ*nfsA*: TTTGCTCATGCTTCCCGCTG, Δ*gshA*: GATT TTGACAGGCGGGAGGT, Δ*sodA*: GCCGTTGTCGATTTACTGGC, and Δ*gshB*: ACCGCGCTACAAGTACGATT. Knockout strains were complemented with plasmids from the unpublished TransBac library (Otsuka *et al*, 2015) by transformation (Fig EV5E). For the Δ*nfsA* and the Δ*gshA* strains, the kanamycin resistance cassette was removed using the plasmid pCP20, as described (Cherepanov & Wackernagel, 1995). All experiments were performed at 30°C in minimal M9 medium (1× M9 salts, 2 mM MgSO$_4$, 0.1 mM CaCl$_2$, supplemented with 4 g/l glucose and 0.1% amicase, pH ~ 7.1). For the experiment shown in Fig 5D, inosine was added at 0.3 mM. Antibiotics for dynamic measurements were dissolved in ethanol [trimethoprim (catalog# 92131), tetracycline (268054)], or dimethylformamide [nitrofurantoin (N7878)] and added from concentrated stocks (stored at −20°C in the dark) at the indicated concentrations. The final concentrations of solvent in the dynamic experiments (0.05% for ethanol,

and 0.04% for dimethylformamide) did not have any effect on cell growth nor gene expression. Antibiotics for selection and glycerol stocks were kanamycin (catalog# K4000) used at 25 μg/ml, ampicillin (A9518) used at 50 μg/ml, spectinomycin (PHR1441) used at 100 μg/ml, all dissolved in water, and chloramphenicol (C0378) used at 10 μg/ml and dissolved in ethanol. All chemicals were obtained from Sigma-Aldrich except when stated otherwise.

**Construction of strains with chromosomally integrated pairs of promoter–fluorescent protein (FP) reporters**

The cloning method we used was optimized for the integration of multiple promoters from a promoter-GFP library (Zaslaver *et al*, 2006) into fixed chromosomal loci, switching fluorescent proteins, and antibiotic resistance markers. Although the chromosomal context may generally affect gene expression (Bryant *et al*, 2014), the validity of our approach is supported by our previous results showing that expression from a chromosomally integrated promoter-YFP construct (*gadB*-YFP) highly correlated with a cellular phenotype, i.e., cell death under acid stress (Mitosch *et al*, 2017). By dealing with short PCR products, this method additionally circumvents the problem that integration of constructs using recombineering gets more difficult with longer insert size (Kuhlman & Cox, 2010). Other advantages include that the long primers typically used in recombineering (Datsenko & Wanner, 2000) are not needed and the same two primer pairs for YFP and CFP can be used for all promoters. This method allowed reliable chromosomal integration with > 50% correct colonies. We first constructed so called "platforms" and integrated them into the chromosome (Fig 3B). These platforms can later accept short PCR products from the library promoters (Fig 3A). The platforms were integrated into two positions opposite and approximately equidistant from the *E. coli* chromosomal origin: *intS* (chromosomal position 2,466,545 -> 2,467,702; Keseler *et al*, 2017) and *galK* (chromosomal position 788,831 <- 789,979 (Keseler *et al*, 2017), Fig 3B). Expression from these positions has been probed before and did not influence bacterial fitness under our experimental conditions. We used the YFP variant Venus and the CFP variant Cerulean (Cox *et al*, 2010) for the dual-reporter method due to their similar and short maturation times ∼ 10 min (Balleza *et al*, 2017) and their good spectral separation that allowed imaging without any corrections for bleedthrough.

For the initial construction of the platforms on plasmids, the origin of replication of the promoterless plasmid from Zaslaver *et al* (2006) was first changed to a pZA origin. The exchange of the inherent GFPmut2 to either a YFP or a CFP (Cox *et al*, 2010) was achieved by using the primers cYFP-1 and cYFP-2 for YFP, and cCFP-1 and cCFP-2 for CFP (see Table EV4 for primer sequences) and the restriction enzymes HindIII and NdeI. In the plasmid with CFP, the KAN resistance cassette was exchanged by an AMP resistance cassette using restriction and ligation. This resulted in the plasmids p-KAN-YFP and p-AMP-CFP. To add a selectable marker for the platform that would be knocked out upon successful integration of the final reporting construct, the CHL resistance cassette from a plasmid library was put between the XhoI and BamHI restriction sites with primers CmR-1 and CmR-2 and cloned into the plasmid p-KAN-YFP, resulting in the plasmid p-KAN-CHL-YFP. Likewise, a KAN resistance cassette was cloned between the XhoI

and BamHI restriction sites with primers KanR-1 and KanR-2, resulting in the plasmid p-AMP-KAN-CFP.

To replace the intact KAN resistance cassette in p-KAN-CHL-YFP by a defunct fragment, starting after the start codon of its protein coding region, whole-plasmid PCR was used with primers CmR-1 and KanF, which also contains an XhoI restriction site. Likewise, to replace the intact AMP resistance in p-AMP-KAN-CFP with a defunct fragment, the same procedure was applied using the primers AmpF and CmEnd. The platforms were integrated into two different chromosomal locations, knocking out the *galK* and *intS* genes (Fig 3B) using lambda-red-recombineering as described in Datsenko and Wanner (2000), with the recombineering plasmid pSIM6 and primers intS-1 and intS-2, or galK-1 and galK-2. Finally, all integrated platforms were checked for mutations by sequencing the PCR product obtained by using primers intS-up, intS-dn, or galK-up, galK-dn on the chromosomal DNA. For the integration of promoters of interest into the platforms, promoters and the necessary homology regions were amplified via PCR and the primers MKan-1 and mYFP for the platform with YFP, and the primers AmpF2 and mCFP for the platform with CFP. Integration was done using recombineering with the recombineering plasmid pSIM19 (Datta *et al*, 2006) and PCR-checked by sequencing the PCR product obtained by using primers intS-up, intS-dn, or galK-up, galK-dn on the chromosomal DNA with the sequencing primers mKanProm_PromSeq and mAmp-Prom_PromSeq (Table EV4).

For the construction of dual-reporter strains, first a promoter of interest was integrated at the *galK* locus, combining with *cfp*. Promoters integrated at this location needed to have a high basal expression of fluorescent protein, as the CFP autofluorescence (Monici, 2005; Mihalcescu *et al*, 2015) would otherwise mask expression changes. Subsequently, the second promoter of interest was integrated at the *intS* locus, combining with *yfp*. Autofluorescence in the YFP channel was negligible. Therefore, also promoters with lower expression could be integrated at this locus. All strains used for dynamic single-cell measurements were transformed with the plasmid pZS41mCherry used for image segmentation. This plasmid was cloned from the plasmid containing the constitutive $P_{LtetO-1}$ promoter with absent Tet repressor (Lutz & Bujard, 1997) and the plasmid pZS2-123 (Cox *et al*, 2010) which contains the fluorescent protein mCherry. Due to the dynamically increasing CFP autofluorescence under NIT stress, we changed *recA-cfp* to *recA-mCherry* as a control in the strain combined with *ybjC*-YFP (Fig EV4C). This was achieved by exchanging the sequence for GFP by the sequence for mCherry on the original library plasmid using the primers mCh1 and mCh2 and the restriction enzymes XhoI and AvrII. The promoter-mCherry sequence was then integrated into the *galK* locus replacing *kanR-cfp* of the CFP platform using the primers AmpF2 and T1_XFP. For the control experiment where YFP and mCherry were used for measuring dynamic gene expression changes (Fig EV4C), we used the plasmid pZS41mCerulean for segmentation. This plasmid was constructed by replacing mCherry by mCerulean in the plasmid pZS41mCherry by restriction digest and ligation. The fluorescent protein mVenus is known to be particularly susceptible to photobleaching (Shaner *et al*, 2005). Although we tried to keep photobleaching to a minimum by keeping light intensities and exposure times low, photobleaching was not completely unavoidable. The highest detected bleaching for all our conditions was 3% per

frame for YFP and 0.5% per frame for CFP, determined by imaging a microcolony with 10-s time interval. In an experiment with 40 frames per imaging position and 3% bleaching per frame, the initially present fluorescence would drop to 30% of its value by the end of the experiment. As under all our conditions, expression from promoters in the YFP channel was strongly increasing over time, we considered this bleaching to have a rather minor effect and did not correct for it.

Our cloning method allowed for efficient chromosomal integration of library promoters and most promoters of interest (*Pois*) could be integrated easily (integrated promoters and their sequences in Table EV5), except for some promoters which were hard to integrate (~ 10%; possibly promoters which have sequence homologies elsewhere in the chromosome). We successfully integrated 23 promoters and five promoter pairs (*ybjC-recA, fpr-recA, recA-gadB, gadB-gadB, and LlacO-1-LlacO-1*; the first promoter is combined with *yfp*, the second one with *cfp*, respectively). All integrands were checked by sequencing the promoter in a PCR product from the *intS* and *galK* regions. In addition, all newly constructed strains were checked for their growth rate using a robotic system with a plate reader (Tecan infinite 500; Chevereau & Bollenbach, 2015), fluorescent protein expression, and resistance to KAN and AMP, respectively. The ratio between the GFP concentration expressed from the plasmid and the YFP expression expressed from the chromosomally integrated copy, measured at the population level using the robotic system with the plate reader, was not exactly the same for all promoters and varied between 2 and 10. As the ratio was not systematically related to the promoter strength, this promoter-specific expression ratio might be explained by the titration of transcription factors by plasmid-borne promoters (Brewster *et al*, 2014) or might have other yet unidentified reasons. Only strains with consistent results in the previously mentioned checks were used for data acquisition. Overall, this method enabled the efficient integration of individual and pairs of promoter-FP constructs into the chromosome, allowing to change antibiotic resistance and FP in one step. With suitable platforms, any pair of resistance marker and reporter protein can be used similarly.

### Population-level data

The population-level data presented in Figs 1A, 4A and 5A, and EV2H are taken from Mitosch *et al* (2017) and were obtained using a robotic liquid handling system and a plate reader as described (Mitosch *et al*, 2017). In Fig 1A, the expression of regulated promoters on a linear scale was normalized between 0 and 1 and averaged. For the oxidative stress response, the responses of the following SoxS and OxyR regulated promoters were averaged: *ahpC, grxA, ybjC, fpr, inaA, marR, and soxS*. For the SOS response, the responses of the following LexA-regulated promoters were averaged: *dinG, ftsK, lexA, polB, recA, ruvA, and uvrD*.

### Microfluidics and time-lapse microscopy

For all microscopy experiments, we used a microfluidics device in which bacteria grow in microcolonies. This device allows switching between different inlets, and equilibration to the new condition happens within minutes (CellASIC ONIX, Merck Millipore). Bacteria were inoculated from frozen glycerol stocks at a dilution of

1:1,000–1:5,000 and grown to an optical density ($OD_{600}$) of 0.05–0.1. Then, they were diluted 1:100 and loaded into the microfluidics chamber, preheated to 30°C. This normally led to spatially well separated single cells in the microfluidics chamber. All experiments were performed in a heated chamber at 30°C. Data acquisition was started after 1–2 h. Images were taken every 10–20 min using a 100× oil objective with an EMCCD camera (Hamamatsu) on a Nikon Eclipse Ti-E with an LED light engine (Lumencor). Excitation and emission filter wavelengths for YFP were CWL/FWHM 513/17 nm and dichroic LP 520 nm, CWL/BW 542/27 nm. Excitation and emission for CFP were 438/24 nm, LP 458 nm, 483/32 nm. Excitation and emission for mCherry were 575/25 nm, LP 596 nm, 641/75 nm. Exposure times were adjusted to the expression level of the respective promoter. Therefore, the expression levels provided in the source data are not directly comparable between different promoters.

Maturation times of CFP and YFP were below 10 min in our conditions, measured by fluorophore accumulation after translational inhibition with chloramphenicol in IPTG-inducible $P_{LlacO-1}$-fluorophore strains (Lutz & Bujard, 1997), as described (Megerle *et al*, 2008), and consistent with results from Balleza *et al* (2017). In contrast, mCherry had a longer maturation time (~ 40 min) and was therefore mostly used as a segmentation color (Figs 4B and 5B), controlled by the constitutive $P_{LtetO-1}$ promoter with absent Tet repressor (Lutz & Bujard, 1997) on a plasmid (pZS41mCherry). Fluorescent protein sequences of *yfp, cfp*, and *mCherry* are from Cox *et al* (2010). Image acquisition was done for ≥ 8 h after the addition of antibiotics.

### Analysis of single-cell data

Microscope movies were segmented and analyzed using an adapted version of the MATLAB program "SchnitzCells" (Young *et al*, 2012). This program does automatic segmentation and tracking but needs manual corrections. At later time points, segmentation and tracking became really difficult: Often the cells got slightly out of focus or the manual corrections became very laborious due to the high number of cells present in a frame. The movie of each microcolony was therefore truncated at a different time point. However, this truncation did not influence our response time determination since expression levels were already saturated at the time point at which each movie was cut. Note that for the IPTG-inducible *LlacO-1* promoter (Fig 2E), expression levels were not saturated yet when we cut the movie at 2 h after induction. Fluorescence background of the surrounding environment was subtracted as the median fluorescence over all pixels outside bacteria. Expression level was determined by dividing the total fluorescence signal from a cell by its total area. Autofluorescence was subtracted using cells devoid of the respective fluorescent protein which were always imaged in the same experiment as the strain of interest. In cases where the autofluorescence background had an upward trend during our dynamic experiments (which was the case for CFP under NIT), the autofluorescence background at each time point was subtracted as the mean expression of cells without that fluorescent protein present at that time point. The clear temporal order and response time correlations for *ybjC* and *recA* under NIT stress did, however, not critically depend on this dynamic background subtraction for CFP and still held when a constant background value was subtracted (Fig EV4B). When *ybjC*-CFP was exchanged for *ybjC*-mCherry as a

further control, the clear temporal order was still present, and the positive response time correlation was still significant albeit weaker (Fig EV4C). This may be explained by the longer maturation time of mCherry (Balleza *et al*, 2017) which may blur correlations. The clear temporal order and response time correlations were also present in an *nfsA* knockout strain (Fig EV5B), suggesting that the potential positive feedback loop via SoxS (Fig EV5A) does not impair this causal chain of molecular events under NIT stress.

Response times were determined as the time point when an expression threshold was reached. This threshold was defined separately for each microcolony as 25% of the maximum expression (minus the minimum expression after stress addition) of the median over all cells with a > 3-fold expression change (Fig 2B). The full median expression and the defined threshold did not vary excessively among different microcolonies of the same promoter and condition (Fig EV6). For some promoters and stress conditions, not 100% of the cells reached the defined threshold (TMP: *recA*: 96%, *fpr*: 85%, *gadW*: 75%, *gadA*: 93%, *ldhA*: 96%, *osmC*: 97%, *wrbA*: 80%; TET: *nrdH*: 90%; NIT: *recA*: 99%). For some promoters and stress conditions [*fpr* (TMP), *wrbA* (TMP), *nrdH* (TET)], these non-responding cells had slightly, but significantly different mean growth rates after stress addition (see Table EV3), suggesting a potential role of the corresponding genes in adaptation to the stressor. Cells that did not reach the threshold were excluded from the analysis. An alternative response time measure (used in Fig EV2C–G) is given by the time until the half-maximum expression level is reached for each single cell. As maximum expression level, the maximum point in each curve was used. This alternative response time measure did not affect our overall conclusions, and the randomness parameter $(\sigma_t/\mu_t)^2$ was highly correlated for the response times determined in both ways, and when time was corrected for the specific single-cell growth rate (Fig EV2C–G). In cases where microcolony size had an impact on gene expression (as for promoters from the Gad system, including *gadB*), data were cropped before this trend became apparent. Note that in Fig 5, data are cropped at 8 h after stress addition for better visibility and comparability with Fig 4. At this time point, expression levels for *recA* were not fully saturated yet in the population data (Fig 5A). For the determination of single-cell response times, we used uncropped data (i.e., until 10–13 h after stress addition). The tree structure of our data (each cell doubles and divides and will ultimately give rise to many progeny cells) reduces variability and overestimates statistical significance when every single cell present in the last imaging frame is included in the analysis. We therefore performed subsampling, where we only followed random single trajectories of each cell present at the time point of stress addition and repeated this procedure many times. Means in Fig 2C (*x*-axis) are the average over the means of 1,000 random samples drawn in this way. Standard deviations in Fig 2C (*y*-axis) are the average over the standard deviations of the same 1,000 random samples. Error bars for means and standard deviations in Fig 2C are from bootstrapping in these 1,000 random samples using the MATLAB function *bootstrp*. Correlation coefficients in Figs 3D, 4C and 5C, and EV3–EV5 are the average over the Pearson correlation coefficients of 1,000 random samples, and the standard deviation of the correlation coefficient is the standard deviation of the same 1,000 random samples. The *P*-value is the median over all *P*-values of the same

1,000 samples. The standard deviation and *P*-value in Fig EV2H were obtained by bootstrapping on 1,000 samples with the MATLAB function *bootstrp*. The *P*-values for the temporal order were obtained by 1,000 times randomly combining *ybjC* response times with *recA* response times and counting how often an equally strict temporal order as the real experiment was observed. The final *P*-value is the mean over all random 1,000 subsamples to which this procedure was applied. Note that we detected slight positive correlations (Pearson correlation coefficient < 0.5) between gene expression and the distance of a cell from the edge of a microcolony. This is explained by the fact that cells in the middle of a microcolony receive more signal from neighboring cells. As response times of single cells are captured at an early stage of the microcolony (at 25% of the median full expression level) and our response time measure is independent of the maximal expression level of a single cell, this effect has a negligible impact on the response times. Single-cell growth rates (Fig EV2A) were determined as the difference between the logarithmic cell lengths between frames, divided by the time interval. The doubling rates of all single cells that were present in the final image of the time series were determined from the cell size at each time point. To this end, we first log2 transformed the actual cell areas. At a cell division, however, this value drops approximately by a factor of 2. In order to have a continuous curve from which we could deduce cell doublings, we extrapolated the log2 value right after cell division using a linear fit through the last few points. This continuous curve was shifted so that the time point at stress addition was set to 0 doublings. Both, the single-cell growth rate and the single cell doubling measure were smoothed with a moving average filter of window size 3 and 5, respectively. The dashed line in Fig 2C was drawn through [0,0] and the lowest $\sigma_t/\mu_t$, which we measured for the *dnaK* promoter.

## Interpretation of gene expression timing using statistical kinetics

The theory of stochastic processes lets us extract valuable information about underlying processes from distributions of aggregate measures. For our purposes here, we are interested in the mean and variance of response times (often called "completion times" in statistical kinetics) and how it can inform us about the number of sequential steps that are necessary to elicit a measurable response. For the data in the main text, these are all the rate-limiting molecular steps that happen from the addition of the stress to the fluorescing of a protein. A chain of events unfolding in a linear fashion can be thought of as a system progressing from states to other states (Inset in Fig EV1). Typically, the waiting time until the next state is assumed to be exponentially distributed which is a characteristic for a memory-less process. This assumption is usually treated as equivalent to the fact that the system retains no memory of how long it has been in this state and the probability of exiting it remains constant over time. To use this theory to make predictions for the behavior of complex systems from the measurements of the completion times, we can look at the distribution of these times. Of central importance is the so called randomness parameter *R* (Moffitt & Bustamante, 2014):

$$R = \left(\frac{\sigma_t}{\mu_t}\right)^2$$

This measure has the intriguing property that it can be used to put a limit on the number of rate-limiting steps $n$ in a process:

$$R^{-1} = n_{\min} \leq n$$

Intuitively, this can be understood by imagining one slow reaction and contrasting this with many fast reactions in a chain that add up to the same mean duration of a process. While in the first case, the completion time comes with the full uncertainty of the exponential distribution ($R = 1$), in the second case the deviations around mean from the individual reactions will typically cancel out to some degree, ultimately leading to a smaller variation in completion time (Fig EV1). Therefore, given similar mean completion times, a process with higher timing precision is an indication of more underlying molecular events. It is unknown whether a linear chain of events governs the promoters in our study. If this were the case, and all events in the chain were equally fast, the inequality above would become an equality and we could thus directly infer the number of rate-limiting steps. However, even if the gene expression responses we investigate involve many more steps and include reversible reactions, branching, or feedback loops, we can interpret $n_{\min}$ as an estimate for the effective number of rate-limiting steps (Moffitt & Bustamante, 2014).

The statistical kinetics model we use assumes that every molecular step is described by a Poisson process, which has an exponential waiting time distribution. For a chain of such molecular steps, the completion times follow an Erlang distribution (a special case of the gamma distribution, which only allows integers as shape parameters). While we cannot directly observe the single molecular steps, we tested whether our response time distribution to estimate the precision limit (*dnaK* promoter, Fig 2C) was consistent with an Erlang distribution. We fitted a gamma distribution to our data using the MATLAB function *gamfit* (Fig 2B) and obtained a rounded shape parameter of 37, consistent with the 37 steps estimated from $\sigma_t/\mu_t$. We then produced 1,000 simulated data sets by sampling from a gamma distribution with the same shape (rounded to integer) and scale parameters and the same number of cells as in our experimental data. We performed a Kolmogorov–Smirnov test (MATLAB function *kstest2*) for each simulated data set to test the hypothesis that the measured and the simulated data stem from the same continuous distribution. This was the case for 99% of the comparisons, supporting that our experimental data for *dnaK* are consistent with an Erlang distribution. We performed the same procedure for all other promoters and conditions shown in Fig 2C. Among those, 16 out of 25 were consistent with an Erlang distribution in > 95% of all comparisons between measured and simulated data (including *dnaK* under TET stress, *gadB* and *recA* under TMP stress, *recA* under NIT stress). An additional eight promoters were consistent in > 70% of the comparisons (including *ybjC* under NIT stress), and one single promoter (*cysK* under NIT stress) showed consistency with an Erlang distribution in only 32% of all comparisons. The *LlacO-1* promoter (Fig 2E) was consistent with an Erlang distribution in 99% of the comparisons for the concentrations 1 mM and 0.1 mM IPTG, and in 86% of the comparisons for 0.01 mM IPTG. Our data therefore show high overall consistency with an Erlang distribution and support that the statistical kinetics model is well applicable to our data set. Note that stochastic models of gene expression predict that the protein concentrations follow a gamma distribution (Shahrezaei & Swain, 2008; Taniguchi *et al*, 2010). This distribution is mathematically closely related to the Erlang distribution we observed for response times (Fig 2B). While protein concentrations and response times are clearly completely different quantities, the similarity of these distributions could be explained if timing variability simply follows the steady-state protein distribution (Co *et al*, 2017).

**Expanded View** for this article is available online.

## Acknowledgements

We thank Andreas Angermayr, Jordi Garcia-Ojalvo, Andreas Hilfinger, Bor Kavčič, Joachim Krug, and Joel Stavans for critical comments on the manuscript and on the interpretation of the data using statistical kinetics, and the Typas and Bollenbach groups for fruitful discussions. We thank Nassos Typas for helpful discussions and access to resources. We thank the imaging facility at IST Austria and Dirk Scholz for technical support. This work was supported in part by Austrian Science Fund (FWF) standalone grant P 27201-B22, HFSP program Grant No. RGP0042/2013, and German Research Foundation (DFG) Collaborative Research Centre (SFB) 1310.

## Author contributions

KM and TB conceived the study and designed the experiments. KM performed the experiments and analyzed the data. TB proposed to interpret the data using statistical kinetics, and GR advised this interpretation. GR conceived the chromosomal integration method and GR and KM constructed chromosomal integration strains. KM, GR, and TB wrote the manuscript.

## Conflict of interest

The authors declare that they have no conflict of interest.

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
