## [Review Process File · Molecular Systems Biology]

Temporal order and precision of complex stress responses in individual bacteria

Karin Mitosch, Georg Rieckh, and Tobias Bollenbach

Review timeline:

Submission date:	28 May 2018
Editorial Decision:	23 July 2018
Revision received:	19 October 2018
Editorial Decision:	30 November 2018
Revision received:	28 December 2018
Accepted:	22 January 2019

Editor: Maria Polychronidou

Transaction Report:

1st Editorial Decision

23 July 2018

Thank you again for submitting your work to Molecular Systems Biology. We have now heard back from the three referees who agreed to evaluate your study. As you will see below, the reviewers think that the presented findings seem potentially interesting. They raise however a series of concerns, which we would ask you to address in a major revision.

Without repeating all the points listed below, one of the more fundamental issues refers to the need to better support and contextualize the results related to the temporal order of events in response to stress. Moreover, reviewer #3 pointed out, also during our pre-decision cross-commenting process in which the reviewers are given the chance to comment on each other's reports, that some level of mechanistic insight would significantly enhance the impact of the study.

All other issues raised by the reviewers would need to be convincingly addressed. As you may already know, our editorial policy allows in principle a single round of major revision so it is essential to provide responses to the reviewers' comments that are as complete as possible. Please feel free to contact me in case you would like to discuss in further detail any of the issues raised by the reviewers.

REFeree REPORTS

Reviewer #1:

The manuscript by Mitosch, Rieckh, and Bollenbach examines the temporal order of gene expression responses due to stress caused by sublethal antibiotic exposure. Using *E. coli* promoters fused to fluorescent reporters, the study measures the response time and its variation with the goal of identifying the temporal ordering in the initiation of gene expression in these stress response pathways. In addition to single-cell level data, the authors develop a model using queuing theory that agrees with the experimental results and identifies upper and lower limits on timing variability. Overall, the manuscript is clearly written and presents a nice narrative about limits on the precision of response time and the temporal ordering of stress response in single cells. There are several points

that could use clarification or additional data.

1. The results in Figs. 4 and 5 appear to be based on a fairly small number of cells (~30). To draw clear conclusions about the temporal order of events (e.g. oxidative stress response strictly preceding SOS response under NIT) the number of cells should be at least an order of magnitude higher. These data can come from different microcolonies in replicate experiments. As an aside: If image segmentation is the rate-limiting step it may be helpful to look into other automated programs beyond Schnitzcells (e.g. SuperSegger or others).
2. It would be helpful to clarify the language surrounding σ . My understanding is that this is the standard deviation of the times to reach the threshold, but it is referred to in the text as variability (confusing for its similarity to variance since "variability" is not explicitly defined) and in the Fig. 2C y axis label refers to σ absolute error.
3. The highly precise identification of the maximum number of rate-limiting steps (37 +/- 9) seems a bit odd. The model does not identify any functional classification and is prone to imprecision at high numbers of steps. I understand that the authors identify this as an upper limit, but would advocate for removing language that overemphasizes a specific number of reaction steps, or adding language that serves as a disclaimer to this precision.
4. The experiments corroborating the predictions about acid and DNA stress in response to TMP using inosine are nice. Is it possible to perform a similar experiment for the SOS and oxidative stress responses in NIT to confirm those results?
5. It would be helpful to add some discussion on what examples of rate-limiting steps are. The model does not specify anything about what they must be, but it would be helpful to have some mechanistic insight into what counts and what does not.
6. Figs. 1A and 4A should have error bars or some other way of representing variability across samples in bulk cultures.
7. Typo: Fig. 3B caption ("ypf")
8. Consider changing away from red-green color pairing to avoid issues for color blind readers.

Reviewer #2:

Summary

In this study, the authors used single-cell analysis to examine the temporal response of various genes in response to stresses.

In the first part of the study, the authors examined the correlation between the response time (μ) of a protein and the variability (σ) in this response. By testing numerous promoters in response to environmental signals (antibiotic treatment), they found an approximately linear correlation between the two. Moreover, their data seem to suggest an upper bound and a lower bound in this correlation.

The authors then suggested that this correlation can be explained by a queuing theory. In this conceptual framework, the expression of each protein can be considered consisting of a series of multiple discrete steps (n). The number of steps would constrain the ratio between the two: $\sigma/\mu > \sqrt{n}$, which sets the precision limit of each response. They suggested that the constraint would enable the estimation of the number of steps in activating a target promoter. For example, applying this theoretic constraint, the authors estimate that the number of steps in the response of these promoters range from 1 to >37. Also, they showed that the same promoter exhibited different ratios when cells were exposed to different stresses. For instance, *recA* is estimated in the 8 steps for NIT stress and 4 steps for TMP stress.

In the second part, the authors developed a dual reporter system to dissect the temporal order of different gene cascades in response to stresses. Specifically, they found that the oxidative stress and SOS responses are in the same cascade; yet the acid stress response and the SOS response are in parallel cascades.

General comments

Overall, I find the work interesting and consisting of extensive well-executed experimental and computational analyses. A caveat is that the work gives the impression of consisting of two parts that are not strongly integrated. Each part is interesting but somewhat lacks depth in analysis. In fact, the work reads like two papers.

To me, the most interesting part is the first part. The key conceptual insight is the use of certain quantitative properties of the cell-cell variability to deduce regulatory structures (in this case, the minimal number of steps involved in each response). The use of noise to deduce biological insights has been carried out previously, this line of research has been under-appreciated and this work provides the demonstration of this concept in a new context. The work consists of extensive, challenging single-cell measurements and solid quantitative analysis. While the theory was previously developed, experimental demonstration of the theory is in itself valuable.

I'm less sure about the importance of establishing the temporal order of two genes (in the same cascade or two cascades). The authors should further clarify the implications of understanding this temporal order. Another limitation is that the second part does appear to strongly build upon the first part. This apparent limitation could potentially be addressed by some textual revisions.

Specific and minor points

1. A limitation of the first part is that the theory predicts a lower bound of variability to mean response time ratio. As such, given a particular value of this ratio, one can only estimate the minimum number of steps involved in activating each gene. It's unclear when the inequality approaches equality, where the constraint has the greatest prediction value. I wish to see the authors elaborate on this point in a revision.

Specifically, I suggest the authors expand the derivation (in the methods or Supplemental) and discuss these issues. For example, strictly speaking, the authors cannot claim that the *recA* promoter has a longer cascade during the response to NIT than during response to TMP. The queuing theory only predicts that $n_{\text{NIT}} > 8$ for the first and $n_{\text{TMP}} > 4$ for the second. Thus, the authors also need to revise the relevant language to make the text more rigorous.

2. Using cell-to-cell variability to dissect signaling networks is a somewhat underappreciated area. In reviewing this part of the literature, the authors should also note some other relevant studies, including Wong et al, *Molecular Cell* 2011 (doi: 10.1016/j.molcel.2011.01.014), which demonstrates the use of noise to dissect regulatory functions in mammalian cells.

3. In figureS2, all chemicals were dissolved in ethanol, thus a control might be needed to exclude the ethanol effect. Also, it will be better to include the explanation of the difference between doubling and time in figure S2, since the upper bound and lower bound seem affected by them.

4. In figureS4, using mcherry showed that the correlation coefficient was different from using original fluorescent choice can be significantly affected. In the explanation, they mentioned the possibility of the longer maturation rate of mcherry. It might be able to screen basal fluorescence protein bias, using the same promoter for both loci with mcherry as the researchers have done in figure S3.

Reviewer #3:

Summary

The group has previously studied the response of *E. coli* to different antibiotic stresses. They observed that genetically identical cells show a quantitatively different response to such stimuli. In the current manuscript, they extend these observations by cloning 23 different transcriptional reporters (derived from the Zaslaver-Alon promoter library) on the chromosome and measuring the time-course of the response to antibiotic addition in single cells using a microfluidics device. The authors focus on analyzing the variability of the timing of induction of the different promoters. They observe large differences between promoters and induction conditions (different antibiotics) and they interpret the observed variability in the context of a model from queuing theory. The analysis of the timing suggests that the oxidative stress and SOS response to nitrofurantoin (NIT) are part of the same pathway, whereas the acid stress and SOS response to trimethoprin (TMP) are mediated by largely independent, parallel pathways.

General remarks

The major scientific question concerns the connection between the timing of the response to a

stimulus (antibiotic stress) and the network (sequence of events) that connects the stimulus to the observed response (expression of a *gfp* reporter gene). The experimental procedures are appropriate for the questions asked and the experiments are well carried out. I do not detect any technical problems.

The authors draw two main types of conclusions from the data: (i) common steps (or not) of regulatory pathways leading to the activation of different promoters following a defined stimulus. (ii) Number of rate-limiting molecular events between the stimulus and the expression change of the reporter gene. While I am rather convinced of the first type of conclusions (even though a correlation only provides strong suggestions, no prove, as the authors correctly point out in the first paragraph of the discussion), I am much more skeptical about the second type of conclusions.

The first type of conclusions concern the ordering of events in response to a stimulus. The authors show that the oxidative stress response strictly precedes the SOS response following NIT addition at the single-cell level. This strongly suggests, but does not prove, a common first part of the pathway leading to the two events. The authors suggest that oxidative stress "causes" subsequent DNA damage, and therefore the SOS response. On the contrary, the response to TMP addition leads to an acid stress and SOS response that do not obey a strict ordering at the single-cell level. This observations does indeed suggest that TMP triggers two different, largely independent chains of events leading to the two distinct stress responses. The main idea of this part is that the observation of the temporal order of gene expression can be used to deduce the sequential steps of a pathway. In itself, this reasoning is not very original. The added value of the manuscript is to perform these experiments at the single-cell level.

The conclusions about the number of molecular events between the stimulus and the expression of the reporter gene critically hinge on the model from queuing theory. This model assumes a linear chain of irreversible events: one stimulus leads to the expression of one *gfp* molecule. While the addition of nucleotides to the mRNA are strictly sequential, the model already breaks down if one promoter produces more than one mRNA. Furthermore, if there is any feedback, branching, or other variants of the chain of events, the queuing model fails. This model is quite appropriate for enzymatic reactions (as presented by Moffitt and Bustamante, 2014), but, in my opinion, not adequate for describing gene expression (also see point 6 below).

Therefore, even though the work claims to analyze a fundamental relationship between timing variability and pathway topology (in particular about the number of intervening steps), this conclusion is contingent on a very debatable model. The other conclusions concerning the topology of individual stress responses based on the sequence of events are justified, but remain at the level of correlations without any mechanistic prove. I therefore estimate that the novelty of the results is modest and that there is limited support for the conclusions concerning the number of rate limiting steps in a gene expression pathway.

Major points

1. The variability of the timing of gene expression is analyzed by measuring the time between the addition of the antibiotic and 25% (or 50% in SI) of the median full response. Quite obviously, different cells attain different expression levels and this should be taken into account for measuring timing variability. The response time should measure the time for attaining 25% of the full expression level of each particular cell. The current measure combines timing variability and variability in the "activity of the gene expression machinery" and potentially other things. Figure S2B already gives a hint that taking into account the specific growth rate of the cells reduces timing variability: the plot S2B shows a higher correlation than plot 2C. You should indicate the correlation coefficient in S2B and S2C.
2. It is not clear to me whether the data presented in S2C were analyzed using the maximal expression level of each cell. If this was the case, please describe how you "fit" the maximal expression level to the data and compare the randomness parameter of the genes between the two ways of analyzing the data (possibly in SI).
3. You specifically exclude lineage related cells to estimate variability. Daughter cells may be more correlated in growth rate and you therefore introduce more variability due to growth rate effects. Could you perform an analysis that includes growth rate and maximal expression levels of single cells (a sort of combination of Figures S2B and S2C).
4. You also exclude a certain number of cells from the analysis because they do not show a more than 3-fold expression change. In one case, this concerns one quarter of the cells. How does the analysis change if you include all cells (using individual maximal expression change as suggested in point 1)? Do the excluded cells have a different growth rate?
5. The well-characterized lac promoter shows a very high precision of timing. According to the

model, this implies about 40 rate-limiting steps. This number is conceivable if, for example, every polymerization step of mRNA production is rate-limiting. However, all reporter genes follow the same steps and some reporter genes are expressed with a similar timing (see Figure 2C), but showing much greater variability. How can you argue for much fewer rate-limiting steps in these cases without increasing the response time?

6. One argument for the validity of the model derives from the observation that the response times follow an Erlang distribution (a special case of a gamma distribution). Many years ago it was already shown that a simple two-step model (transcription, translation) of stochastic gene expression leads to a gamma distribution (see, for example, Friedman et al., Phys. Rev. Lett. 97, 168302, 2006 or Shahrezaei & Swain, PNAS, 105, 17256, 2008) of protein concentrations. Furthermore, a more recent publication (Dal Co et al., NAR 45, 1069, 2017) shows that timing variability follows the steady-state protein distribution. You should comment on these alternative, rather simple models and put your proposal into perspective with this literature.

Minor points

7. All primary data (time series for all genes and cells) should be provided in SI.

8. The microfluidics experiments involve the analysis of micro-colonies. Do you observe any effect on expression based on the location of the cell within the colony?

9. You normalize the expression threshold by the median full expression range of a colony. How does this expression range vary between colonies? This question is related to questions 1 above.

1st Revision - authors' response

19 October 2018

Point-by-point response to reviewers

Reviewer #1:

The manuscript by Mitoš, Rieckh, and Bollenbach examines the temporal order of gene expression responses due to stress caused by sublethal antibiotic exposure. Using *E. coli* promoters fused to fluorescent reporters, the study measures the response time and its variation with the goal of identifying the temporal ordering in the initiation of gene expression in these stress response pathways. In addition to single-cell level data, the authors develop a model using queuing theory that agrees with the experimental results and identifies upper and lower limits on timing variability. Overall, the manuscript is clearly written and presents a nice narrative about limits on the precision of response time and the temporal ordering of stress response in single cells. There are several points that could use clarification or additional data.

1. The results in Figs. 4 and 5 appear to be based on a fairly small number of cells (~30). To draw clear conclusions about the temporal order of events (e.g. oxidative stress response strictly preceding SOS response under NIT) the number of cells should be at least an order of magnitude higher. These data can come from different microcolonies in replicate experiments. As an aside: If image segmentation is the rate-limiting step it may be helpful to look into other automated programs beyond Schnitzcells (e.g. SuperSegger or others).

We thank the reviewer for this comment. To show that there is a clear temporal order in gene expression, we agree that this order needs to be observed consistently in a large number of cells. In the previous version of the manuscript, the number of bacterial cells we analyzed for the oxidative stress and DNA stress (Figure 4) was not clearly stated. We analyzed at least two microcolonies in each of three replicate experiments and we have now included additional microcolonies in the revised manuscript. Adding up all cells, we analyzed 83 cells for the wild type (see Table EV2). In addition, there are another 59 cells for the *ybjC*-YFP/*recA*-mCherry combination: In none of these cells did *recA* precede *ybjC*. Here, it is important to emphasize that we only include cells that are present at the time of stress addition, when the microcolonies are still relatively small, in this count. We have added a permutation test (page 7, paragraph 1) in which we randomly combine the observed *ybjC* response times with the *recA* response times and count in how many cases we would observe a similarly strict temporal order as observed in the experiment. This test supports that the observed strict temporal order is significant ($p=0.017$) and the number of cells we observed is thus sufficient. Moreover, the high positive correlations between the response times for *ybjC* and *recA* strongly support a causal relationship between the two responses.

Note that for the experiment about acid stress and DNA stress response under TMP (Figure 5), no clear temporal order is observed. In this case, fewer cells suffice to support this result: It is already

clear from a smaller number of cells that *gadB* can precede *recA* or *vice versa* in different cells (Figure 5C).

In summary, we have analyzed more cells and obtained consistent results, thus strengthening our conclusions. We have further clarified the number of bacterial cells analyzed in Table EV2. In addition, we added a statistical test to exclude the possibility that the observed order of the response times is due to random chance and too few observed cells (page 7, paragraph 1). We also revised the description of Figure 5C to clarify that a considerable number of cells violate the population-level temporal order of *gadB* and *recA* under TMP (page 7, paragraph 4).

2. It would be helpful to clarify the language surrounding σ . My understanding is that this is the standard deviation of the times to reach the threshold, but it is referred to in the text as variability (confusing for its similarity to variance since "variability" is not explicitly defined) and in the Fig. 2C y axis label refers to σ absolute error.

We thank the reviewer for pointing this out. In the revised manuscript, we have clarified the language and refer to σ as the standard deviation of the response times (page 3, paragraphs 3, 4; page 4, paragraph 1, 2). We also changed the y-axis label in Fig. 2C accordingly.

3. The highly precise identification of the maximum number of rate-limiting steps (37 +/- 9) seems a bit odd. The model does not identify any functional classification and is prone to imprecision at high numbers of steps. I understand that the authors identify this as an upper limit, but would advocate for removing language that overemphasizes a specific number of reaction steps, or adding language that serves as a disclaimer to this precision.

We agree with the reviewer. We have included a disclaimer in the text that it is hard to estimate this number precisely from the data (page 5, paragraph 2), and we now refer to it as "~37" steps throughout the text.

4. The experiments corroborating the predictions about acid and DNA stress in response to TMP using inosine are nice. Is it possible to perform a similar experiment for the SOS and oxidative stress responses in NIT to confirm those results?

In contrast to the TMP experiment, where we could simply change the growth medium to abolish the acid stress, we could not find a similar straightforward condition for altering the cellular response to NIT. We still added a new experiment that uses genetic manipulations aimed at altering the oxidative stress caused by NIT. We selected knockout mutants that affect the cellular ability to counteract oxidative stress and tested their growth rate under NIT stress. We identified three mutants (Δ *sodA*, Δ *gshA*, Δ *gshB*) with clear growth rate defects under NIT stress; these could be rescued by complementation with the respective gene (Figure EV5D,E). We further tested the gene expression of *ybjC*-YFP and *recA*-CFP in the *gshA* knockout strain in response to NIT and observed a stronger oxidative stress response followed by a stronger DNA stress response in this mutant compared to the wild type. These results corroborate the causal role of oxidative stress in DNA damage under NIT exposure. We have included four new figure panels (Figure EV5D-G) depicting these results and a corresponding description in the main text (page 7, paragraph 3).

5. It would be helpful to add some discussion on what examples of rate-limiting steps are. The model does not specify anything about what they must be, but it would be helpful to have some mechanistic insight into what counts and what does not.

We thank the reviewer for pointing this out. We added several sentences to discuss what could be plausible rate-limiting steps. We also added references to the literature describing several specific steps that could be rate-limiting (page 5, paragraph 2 and page 8, paragraph 4). In the revised manuscript, we also used our data to estimate the time scale on which such rate-limiting steps occur: For the earliest and most precise promoters, we estimate a duration of 1-3 min (page 5, paragraph 2). This estimate clearly shows that single steps in transcription or translation, which are about 1000 times faster, are not rate-limiting here. Transcription initiation or fluorophore maturation may, however, well belong to the rate-limiting steps. In general, a rate-limiting step could also be the entry of the antibiotic into the cell. For TMP, the acidification of the cytosol could occur on a time-scale of minutes and thus be another rate-limiting step. These estimates are now discussed in the results part (page 5, paragraph 2) and in the Discussion (page 8, paragraph 4).

6. Figs. 1A and 4A should have error bars or some other way of representing variability across samples in bulk cultures.

Thanks for pointing this out. In general, these population-level experiments are highly reproducible from day to day for sufficiently strongly expressed promoters. We have now included error bars for the different population-level experiments. In Figure 1A, the error bars are the mean and the standard deviation over the different oxidative and LexA-regulated promoters from one experiment. Under NIT stress, all observed oxidative stress promoters behaved very similarly; the same holds for all LexA-regulated promoters. We note here that such high similarity within one stress response was not generally observed for other stress responses and conditions (Mitosch, 2017). The error bars we added in Figures 4A and 5A are the standard deviation over three replicate measurements done on different days for the respective promoters. We have changed the figure legends accordingly.

7. Typo: Fig. 3B caption ("ypf")

We thank the reviewer for pointing out this typo.

8. Consider changing away from red-green color pairing to avoid issues for color blind readers.

We thank the reviewer for pointing this out. The chosen color scheme has the advantage that in the overlay of the microscopy images (Figure 4B and 5B) the colors result in a third, distinguishable color: red and green add up to yellow, and red and blue to purple. Although we have kept the overall color scheme for that reason, we have adjusted the HSB values of the colors to make the red and green clearly distinguishable for colorblind readers. We validated this property using the Adobe Illustrator proof setup.

Reviewer #2:

Summary

In this study, the authors used single-cell analysis to examine the temporal response of various genes in response to stresses.

In the first part of the study, the authors examined the correlation between the response time (μt) of a protein and the variability (σt) in this response. By testing numerous promoters in response to environmental signals (antibiotic treatment), they found an approximately linear correlation between the two. Moreover, their data seem to suggest an upper bound and a lower bound in this correlation. The authors then suggested that this correlation can be explained by a queuing theory. In this conceptual framework, the expression of each protein can be considered consisting of a series of multiple discrete steps (n). The number of steps would constrain the ratio between the two: $\sigma t / \mu t > \sqrt{n}$, which sets the precision limit of each response. They suggested that the constraint would enable the

estimation of the number of steps in activating a target promoter. For example, applying this theoretic constraint, the authors estimate that the number of steps in the response of these promoters range from 1 to >37 . Also, they showed that the same promoter exhibited different ratios when cells were exposed to different stresses. For instance, *recA* is estimated in the 8 steps for NIT stress and 4 steps for TMP stress.

In the second part, the authors developed a dual reporter system to dissect the temporal order of different gene cascades in response to stresses. Specifically, they found that the oxidative stress and SOS responses are in the same cascade; yet the acid stress response and the SOS response are in parallel cascades.

General comments

Overall, I find the work interesting and consisting of extensive well-executed experimental and computational analyses. A caveat is that the work gives the impression of consisting of two parts that are not strongly integrated. Each part is interesting but somewhat lacks depth in analysis. In fact, the work reads like two papers. To me, the most interesting part is the first part. The key conceptual insight is the use of certain quantitative properties of the cell-cell variability to deduce regulatory structures (in this case, the minimal number of steps involved in each response). The use of noise to deduce biological insights has been carried out previously, this line of research has been under-appreciated and this work provides the demonstration of this concept in a new context. The work consists of extensive, challenging single-cell measurements and solid quantitative analysis. While the theory was previously developed, experimental demonstration of the theory is in itself valuable.

I'm less sure about the importance of establishing the temporal order of two genes (in the same cascade or two cascades). The authors should further clarify the implications of understanding this temporal order.

We thank the reviewer for appreciating our work. We have tried to corroborate both parts of our study by adding two new experiments.

1. In the first new experiment, we quantified the mean and the standard deviation of the response time of the *LlacO-1* promoter induced using different IPTG concentrations (Figure 2, new panel E). Here, the idea is that, by reducing the inducer concentration, the time it takes a fixed number of IPTG molecules to enter the cell and bind the lac repressor becomes longer. Consequently, these steps should at some point become rate-limiting and the overall number of effective rate-limiting steps should decrease (the previously rate-limiting steps are faster in comparison and can no longer contribute to lowering timing variability). This experiment indeed showed that, at low IPTG concentrations, the standard deviation of the response time increases disproportionately strongly with the mean of the response time. Interpreting these data using the model from queuing theory shows that the *LlacO-1* promoter is determined by fewer rate-limiting steps at low IPTG concentrations.

2. In a second approach, we first tested whether oxidative stress knockout mutants have a growth defect under NIT compared to the wild type. Three oxidative stress mutants ($\Delta sodA$, $\Delta gshA$, $\Delta gshB$) had a clear and specific growth rate defect under NIT; this defect was complemented by adding back the respective genes. In addition, we tested the expression of *ybjC*-YFP and *recA*-CFP in the $\Delta gshA$ mutant under NIT and found a stronger oxidative stress response, followed by a stronger DNA stress response, compared to the wild type in the same conditions (Figure EV5D-G). These results further corroborate the important role of oxidative stress under NIT exposure and its likely causal role for downstream DNA damage.

We agree with the reviewer that the implications of establishing the temporal order of two promoters need to be more clearly explained. Firstly, our results show that caution must be taken when inferring genetic cascades based on temporal data at the population level: Two genes that appear clearly sequential at the population level may actually not be part of the same regulatory cascade. Secondly, our approach provides a deeper understanding of complex bacterial stress responses. A complete understanding of a bacterial stress response would imply that we know all contributing molecules and interactions between them. In particular, for antibiotics, such an understanding could enable us to modify the bacterial response and prevent undesired outcomes or improve the antibacterial action of compounds. While we are far from achieving a complete understanding of these complex responses, knowing whether two genes belong to the same cascade of events enables us to predict if interfering with one of the cascades affects the other one or not and at which stage one should interfere to achieve the desired outcome. We have added several sentences in the discussion that make this point clearer (page 8, paragraph 3).

Another limitation is that the second part does appear to strongly build upon the first part. This apparent limitation could potentially be addressed by some textual revisions.

Together with the reviewer's comments above that our work "gives the impression of consisting of two parts that are not strongly integrated" and that it "reads like two papers", we understand that we need to make the relation between both parts of our study clearer. Part 1 reports general quantitative trends for the response time variability of individual promoters. This is an important prerequisite for part 2, which focuses on the temporal order of response cascades and response time correlations for multiple promoters in the same cell. The questions addressed in part 2 are naturally motivated by the timing variability observed in part 1.

To clarify this connection and improve the segue between both parts, we added and revised several sentences at the beginning of the sections "Cloning method enables efficient chromosomal integration of promoter pairs" (page 6, paragraph 3) and "The oxidative stress response strictly precedes the SOS response in every single cell under NIT stress" (page 6, paragraph 4).

Specific and minor points

1. A limitation of the first part is that the theory predicts a lower bound of variability to mean response time ratio. As such, given a particular value of this ratio, one can only estimate the

minimum number of steps involved in activating each gene. It's unclear when the inequality approaches equality, where the constraint has the greatest prediction value. I wish to see the authors elaborate on this point in a revision.

We agree that these points need to be elaborated. There are several aspects of the queuing model that are valuable for the interpretation of our data but were not explained in sufficient detail in the previous version of the manuscript.

First, it is correct that, with the result from queuing theory, one can only estimate the minimum number of steps. The inequality becomes an equality for a linear chain of irreversible steps that are all equally fast. There is no doubt that the estimated lower bound is greatly exceeded and there are many more

molecular steps that need to take place before a gene expression response becomes detectable in our experiments: For example, technically, each nucleotide addition in transcription is a molecular step of its own. Thus, it is clear that at least thousands of steps are involved in the responses we study. However, the crucial point is that most of these steps are too fast to affect the timing variability for completion of the entire cascade – only the slowest steps matter for the resulting timing variability. Importantly, the lower bound for the number of steps calculated from our data can be interpreted as an estimate for the effective number of rate-limiting steps (Moffitt and Bustamante, 2014), i.e. the number of relatively slow molecular reaction steps that lie between the addition of the stressor and the detection of the fluorescent reporter signal for a gene expression response. These steps all need to be similarly slow – if one step becomes much slower than the others, that step alone becomes rate-limiting, which leads to increased timing variability. In principle, if a clear separation of time scales between the slowest step and all remaining steps is achieved, the upper bound for timing variability in Figure 2C (dotted line) would be reached. This motivated the new experiment for the *lac* promoter at very low IPTG concentrations which we added; this experiment indeed showed a drastic increase in timing variability at very low IPTG concentrations, corresponding to a reduction in the number of rate-limiting steps (new Figure 2E).

In the revised manuscript, we have clarified this point by adapting Figure 2D, adding the new panel E in Figure 2, and by adding and revising several sentences in the main text to explain this issue better (page 4, paragraphs 2,3).

Specifically, I suggest the authors expand the derivation (in the methods or Supplemental) and discuss these issues. For example, strictly speaking, the authors cannot claim that the *recA* promoter has a longer cascade during the response to NIT than during response to TMP. The queuing theory only predicts that $n_{\text{NIT}} > 8$ for the first and $n_{\text{TMP}} > 4$ for the second. Thus, the authors also need to revise the relevant language to make the text more rigorous.

It is correct that we could only draw the conclusion that n_{TMP} is smaller than n_{NIT} for a linear sequence of steps that all happen at the same rate (where the inequality in Eq. (1) becomes an equality), which is unlikely to be the case. We have revised the text accordingly and now emphasize that (a) relatively few rate-limiting steps are involved in the activation of the SOS response, and (b) the different response time distributions suggest that different upstream processes lead to the activation of *recA* under TMP and NIT, respectively (page 6, paragraph 2).

2. Using cell-to-cell variability to dissect signaling networks is a somewhat underappreciated area. In reviewing this part of the literature, the authors should also note some other relevant studies, including Wong et al, Molecular Cell 2011 (doi: 10.1016/j.molcel.2011.01.014), which demonstrates the use of noise to dissect regulatory functions in mammalian cells.

We agree that this study is relevant in the context of our work and added this citation (page 8, paragraph 3).

3. In figureS2, all chemicals were dissolved in ethanol, thus a control might be needed to exclude the ethanol effect. Also, it will be better to include the explanation of the difference between doubling and time in figure S2, since the upper bound and lower bound seem affected by them. We have performed controls for the ethanol concentration (0.05%) and dimethylformamide (DMF) concentration (0.04%) used in our assays (nitrofurantoin is dissolved in DMF). We could not detect any effect on growth rate in comparison to a control without any ethanol and DMF, and no effect on the expression of *ybjC* under NIT stress, or *gadB* gene expression under TMP stress. We mention this now in the methods section (page 10, paragraph 1).

Concerning the cell doublings in Figure EV2 (former Figure S2): overall, we found that upon correcting for single-cell growth rates, the most precise promoters have a higher standard deviation. This implies that growth rate fluctuations between single cells cannot explain the measured

variability in gene expression. We mention this point now in the main text (page 3, paragraph 4). In addition, we have added a descriptive sentence in the figure legend of figure EV2B, and added a more detailed description of how we obtain these cell doublings in the Methods section (page 19, paragraph 1).

4. In figureS4, using mcherry showed that the correlation coefficient was different from using original fluorescent choice can be significantly affected. In the explanation, they mentioned the possibility of the longer maturation rate of mcherry. It might be able to screen basal fluorescence protein bias, using the same promoter for both loci with mcherry as the researchers have done in figure S3.

We agree that this additional control would be interesting, but we think that it goes beyond the scope of our study and would add relatively little for the cost and effort involved. We used CFP and YFP due to their optimal properties (brightness, maturation time) for our purposes in this experiment and did the relevant control for this combination of fluorescent proteins. We included Figure EV4C as an independent control of our results, using mCherry as another fluorescent protein even though it has less suitable properties. We consider it sufficient that this control experiment qualitatively confirms the results.

Reviewer #3:

Summary

The group has previously studied the response of *E. coli* to different antibiotic stresses. They observed that genetically identical cells show a quantitatively different response to such stimuli. In the current manuscript, they extend these observations by cloning 23 different transcriptional reporters (derived from the Zaslaver-Alon promoter library) on the chromosome and measuring the time-course of the response to antibiotic addition in single cells using a microfluidics device. The authors focus on analyzing the variability of the timing of induction of the different promoters. They observe large differences between promoters and induction conditions (different antibiotics) and they interpret the observed variability in the context of a model from queuing theory. The analysis of the timing suggests that the oxidative stress and SOS response to nitrofurantoin (NIT) are part of the same pathway, whereas the acid stress and SOS response to trimethoprim (TMP) are mediated by largely independent, parallel pathways.

General remarks

The major scientific question concerns the connection between the timing of the response to a stimulus (antibiotic stress) and the network (sequence of events) that connects the stimulus to the observed response (expression of a *gfp* reporter gene). The experimental procedures are appropriate for the questions asked and the experiments are well carried out. I do not detect any technical problems. The authors draw two main types of conclusions from the data: (i) common steps (or not) of regulatory pathways leading to the activation of different promoters following a defined stimulus. (ii) Number of rate-limiting molecular events between the stimulus and the expression change of the reporter gene. While I am rather convinced of the first type of conclusions (even though a correlation only provides strong suggestions, no prove, as the authors correctly point out in the first paragraph of the discussion), I am much more skeptical about the second type of conclusions. The first type of conclusions concern the ordering of events in response to a stimulus. The authors show that the oxidative stress response strictly precedes the SOS response following NIT addition at the single-cell level. This strongly suggests, but does not prove, a common first part of the pathway leading to the two events. The authors suggest that oxidative stress "causes" subsequent DNA damage, and therefore the SOS response. On the contrary, the response to TMP addition leads to an acid stress and SOS response that do not obey a strict ordering at the single-cell level. This observations does indeed suggest that TMP triggers two different, largely independent chains of events leading to the two distinct stress responses. The main idea of this part is that the observation of the temporal order of gene expression can be used to deduce the sequential steps of a pathway. In itself, this reasoning is not very original. The added value of the manuscript is to perform these

experiments at the single-cell level.

We thank the reviewer for appreciating the quality of our experiments and for her/his critical comments. We agree that deducing sequential steps in a pathway from observations of temporal gene expression is an established approach. However, this approach has largely been limited to population-level studies. We are confident that our work provides novel insights and that it makes a strong case for the importance of investigating complex cellular response programs at the single-cell level. In particular, we show that conclusions based on population-level gene expression data can easily be misleading (Figure 5): The expression of the acid stress and the DNA stress promoter under TMP looks perfectly sequential at the population level, but the single-cell level experiment reveals that any conclusion about sequential steps of the pathway is erroneous. A causal role of acid stress in DNA stress can essentially be ruled out – a complete reversal of the conclusion suggested by the population-level data! Hence, performing these experiments at the single-cell level is not a small ploy, but the crucial aspect of this work. Importantly, we also present an example that demonstrates how the classical reasoning can be rescued, and the conclusions about likely sequential steps corroborated by single-cell level data (Figure 4). We are not aware of prior work using this approach (for the study of bacterial stress responses or other systems).

We rephrased several sentences in the main text that explain the conclusions from the experiments in Figure 4 and 5, to more clearly highlight the original aspects that are only observable in single-cell experiments (page 6, paragraph 4; page 7, paragraph 1; page 7, paragraph 4). We hope that this makes the original aspects of our work clearer.

The conclusions about the number of molecular events between the stimulus and the expression of the reporter gene critically hinge on the model from queuing theory. This model assumes a linear chain of irreversible events: one stimulus leads to the expression of one gfp molecule. While the addition of nucleotides to the mRNA are strictly sequential, the model already breaks down if one promoter produces more than one mRNA. Furthermore, if there is any feedback, branching, or other variants of the chain of events, the queuing model fails.

This model is quite appropriate for enzymatic reactions (as presented by Moffitt and Bustamante, 2014), but, in my opinion, not adequate for describing gene expression (also see point 6 below). Therefore, even though the work claims to analyze a fundamental relationship between timing variability and pathway topology (in particular about the number of intervening steps), this conclusion is contingent on a very debatable model. The other conclusions concerning the topology of individual stress responses based on the sequence of events are justified, but remain at the level of correlations without any mechanistic prove.

I therefore estimate that the novelty of the results is modest and that there is limited support for the conclusions concerning the number of rate limiting steps in a gene expression pathway.

We thank the reviewer for this valuable comment that helped us to improve the presentation of this model and the conclusions we draw from it. These concerns prompted us to revisit this model and discuss its application to our data in more detail with two experts for the theory of stochastic processes and its applications in biology (added in the Acknowledgment): Prof. Joachim Krug (University of Cologne) and Prof. Andreas Hilfinger (University of Toronto). There are several misunderstandings here that are largely due to the way we had presented this model in the main text and in the simplified schematic in Figure 2D of the original manuscript.

We introduce the queuing model to provide a context for the observed variability in response timing. The validity of the results from the queuing model as applied to our experimental data indeed relies on assumptions that could be violated (note that this is universally the case for theoretical models in biology). However, the main result in inequality (1) and the corresponding estimate of the lower bound for the number of rate-limiting steps holds generally for Markov processes of arbitrary complexity. In particular, the chain of events does not need to be linear and the events can be reversible; further, feedback loops, branching, or any other variant of the chain (or, rather, "meshwork") of events do not cause any problems: Inequality (1) still holds (see Aldous and Shepp, 1987 for the proof of this theorem). The main assumption that is needed for this result to hold is that each step is memoryless (i.e. it is described as a Poisson process with an exponential waiting time distribution). However, this property is widely accepted as a good approximation for the molecular reaction steps that underlie intracellular processes like those we investigate in this work.

The restrictions highlighted by the reviewer are relevant for the case where the inequality (1) becomes an equality: Indeed, equality in (1) only holds if the sequence of rate-limiting steps is

strictly linear (i.e. no branching or loops), the steps are irreversible, and all steps occur at exactly the same rate. This is certainly not the case for the intracellular processes we investigate: In general, we do not know if the activation of the promoters we measure involves loops, feedback etc., but we can be almost certain that the many events preceding their activation do not all happen at the same rate. Thus, we are left with inequality (1) and can only deduce a lower bound for the number of steps. (The situation is qualitatively the same for the enzymatic reactions, where the queuing model is more commonly applied – the main difference is that the number of reaction steps is much larger in our system.)

Importantly, however, we can interpret this lower bound as an estimate for the effective number of rate-limiting steps (see Moffitt and Bustamante, 2014), i.e. the number of relatively slow molecular reaction steps that lie between the addition of the stressor and the detection of the fluorescent reporter signal for a gene expression response. While the total number of molecular steps is certainly at least in the thousands (e.g. each nucleotide added in transcription is at least one step), it is interesting to estimate how many rate-limiting steps are needed to produce the observed timing variability. These steps all need to be similarly slow – if one step becomes much slower than the others, that step alone becomes rate-limiting, which leads to an increased timing variability. In principle, if a clear separation of time scales between the slowest step and all remaining steps is achieved, the upper limit for timing variability in Figure 2C would be reached. These considerations motivated the new experiment for the lac promoter at very low IPTG concentrations we added; this experiment indeed showed a disproportionate, drastic increase in timing variability at very low IPTG concentrations, corresponding to a reduction in the number of rate-limiting steps (new Figure 2E; see also reply to specific point 1 of reviewer #2). This further supports that the model from queuing theory provides a useful context for our experimental observations.

To clarify the underlying assumptions of the model from queuing theory, the validity of Eq. (1), and the interpretation of the estimated number of rate-limiting steps, we have changed Figure 2D, added the new panel E in Figure 2, and revised the text (page 4, paragraph 2) to emphasize that the studied promoters could also deviate from a strictly linear activation sequence. Since the term “queuing” itself may suggest a linear sequence of events, we also changed “queuing theory” to “statistical kinetics” throughout the manuscript – this closely related term is commonly used in studies that apply these theoretical results to biological systems (see e.g. Moffitt and Bustamante, 2014).

We also agree with the reviewer that mechanistic evidence is valuable for corroborating the scenarios supported by the single-cell gene expression data. The absence of a causal link between acid stress and DNA stress is supported by an experiment where the acid stress response is abolished by adding inosine to the growth medium (Figure 5D): DNA stress occurs without preceding acid stress, all but ruling out a mechanism where acid stress is required for causing DNA stress. To also strengthen our conclusions about the importance of oxidative stress for the downstream damage under NIT with mechanistic evidence, we tested several oxidative stress knockout mutants for their growth defect under NIT stress and found that three mutants ($\Delta sodA$, $\Delta gshA$, $\Delta gshB$) had a severe growth defect under NIT which could be rescued by complementation (Figure EV5D,E). In addition, we tested the expression of *ybjC*-YFP and *recA*-CFP under NIT stress and found a stronger oxidative stress response followed by a stronger DNA stress response compared to the wild type (Figure EV5F,G), supporting a causal role of oxidative stress in DNA stress.

Major points

1. The variability of the timing of gene expression is analyzed by measuring the time between the addition of the antibiotic and 25% (or 50% in SI) of the median full response. Quite obviously, different cells attain different expression levels and this should be taken into account for measuring timing variability. The response time should measure the time for attaining 25% of the full expression level of each particular cell. The current measure combines timing variability and variability in the "activity of the gene expression machinery" and potentially other things. We agree that the response time until a relative fraction of the maximum expression level of a cell is reached can be an informative measure. Thus, we also performed this analysis by defining the response time as the time until 50% of the full expression level is reached for each individual cell (Figure EV2D). This did not change any of our conclusions. We added a sentence to explain this in the main text (page 4, paragraph 1). Ultimately, the exact definition of the response time is largely arbitrary and the different possibilities

(same threshold across cells or relative threshold adjusted for the maximum expression level of each cell) have their own advantages and disadvantages. Using the same (relatively low) threshold for all cells enables us to detect how long it takes the cells to respond to the stress (some threshold is needed to avoid that gene expression fluctuations blur the response time measurement). Using a threshold that is adjusted for the maximum expression level of each cell is a better measure for the time it takes the cells to reach the new gene expression state (in which they have adapted to the stress). While we mention both approaches, we decided to use primarily the same threshold for all cells since it measures the earliest detectable response to the stress and is thus more relevant for our analysis. In addition, a response time measure that is independent of the maximal expression level of an individual cell is more robust to potential errors in segmentation, which are more likely to occur at later time points.

Figure S2B already gives a hint that taking into account the specific growth rate of the cells reduces timing variability: the plot S2B shows a higher correlation than plot 2C. You should indicate the correlation coefficient in S2B and S2C.

We have added the correlation coefficients in these figures. We found that taking into account the specific growth rate of each individual cell does not lead to any major changes in the timing variability. However, timing variability is actually increased for the highly precise promoters. This may be due to technical limitations as it is hard to measure the single-cell growth rate accurately and taking it into account could therefore introduce additional variability. However, this result indicates that the single-cell growth rate is not the main cause of the observed timing variability.

In the revised manuscript, we now explicitly mention this conclusion and the analysis of the effects of growth rate on timing variability (page 3, paragraph 4; page 19, paragraph 1).

2. It is not clear to me whether the data presented in S2C were analyzed using the maximal expression level of each cell. If this was the case, please describe how you "fit" the maximal expression level to the data and compare the randomness parameter of the genes between the two ways of analyzing the data (possibly in SI).

In Figure EV2D (former Figure S2C), we indeed define response time as the time until the half-maximal expression level is reached for each individual cell. The maximal expression level of each promoter was taken literally as the maximal value of its expression without any fit (this value is a good approximation as fluctuations are relatively small at high expression levels; see curves in Figures 4B, 5B).

In the revised manuscript, we explain this better in the methods section (page 18, paragraph 2). For clarification, we also added Figure EV2C, which depicts how the response time was determined for Figure EV2D. We also compare the randomness parameter for both response time measures (Figure EV2F) and for both response time measures after correcting for the specific single-cell growth rate (Figure EV2G).

3. You specifically exclude lineage related cells to estimate variability. Daughter cells may be more correlated in growth rate and you therefore introduce more variability due to growth rate effects. Could you perform an analysis that includes growth rate and maximal expression levels of single cells (a sort of combination of Figures S2B and S2C).

We performed this analysis and added a corresponding plot to the extended view (Figure EV2E). Interestingly, taking into account growth rate effects introduces more variability for highly precise promoters. This may be a technical problem, as it is hard to determine single-cell growth rates accurately, but it certainly suggests that the single-cell growth rate is not a major source of variability in the response time data (see also response to point 1 above). We have added a statement about this in the main text (page 3, paragraph 4).

4. You also exclude a certain number of cells from the analysis because they do not show a more than 3-fold expression change. In one case, this concerns one quarter of the cells. How does the analysis change if you include all cells (using individual maximal expression change as suggested in point 1)?

Do the excluded cells have a different growth rate?

It is correct that we exclude cells that do not cross a certain fold-change in expression for the determination of our threshold. For the analysis, we only exclude cells that do not cross the determined threshold. We checked whether the excluded cells (i.e. cells that do not reach the defined threshold) have different growth rates than the cells that exceed the threshold. Indeed, we found a few marginally significant differences (Table EV3). This suggests that these genes might have

important functions in the bacterial adaptation to the relevant stress. However, this hypothesis would need deeper investigation and goes beyond the scope this work. We now mention these differences in the methods section (page 18, paragraph 2).

As the reviewer correctly points out, not all cells respond for every tested promoter and for some promoters, this number gets high. To avoid introducing artificial noise into our data (non-responding cells would show a response time that is completely determined by noise), we have to use this cutoff. Without this cutoff, we would in essence measure noise, which is why we cannot include such an analysis.

Concerning our dual-color experiments in Figure 4 and Figure 5, we do not see any cases where *recA* crosses the threshold but *ybjC* does not under NIT stress. However, under TMP stress, we see cases in which either *gadB* or *recA* do not cross the threshold. This does typically not happen in the same cells. These finding again support the hypothesis that *ybjC* and *recA* belong to the same molecular chain of events under NIT stress, whereas *gadB* and *recA* belong to different chains of events under TMP stress.

5. The well-characterized lac promoter shows a very high precision of timing. According to the model, this implies about 40 rate-limiting steps. This number is conceivable if, for example, every polymerization step of mRNA production is rate-limiting. However, all reporter genes follow the same steps and some reporter genes are expressed with a similar timing (see Figure 2C), but showing much greater variability. How can you argue for much fewer rate-limiting steps in these cases without increasing the response time?

This is an important point, which we realized needs to be explained in more detail in our manuscript. In brief, gene expression is a process with many (at least thousands of) molecular steps, but only a small fraction of these steps is rate limiting. Fast steps essentially do not affect timing variability. Two promoters can exhibit the same response time but different timing variability in our assay: For example, a promoter that has a single rate-limiting step that takes 30 min on average would have higher variability (but the same average response time) than a reporter that has 10 rate-limiting steps each of which takes 3 min on average.

Now consider specific steps that need to happen before a fluorescence signal from one of the promoters we investigate is detected. In the case of the *LlacO-1* promoter, this starts with the uptake of IPTG, the binding of IPTG to the lac repressor, conformational changes, the binding of the polymerase to the promoter, unwinding of the DNA, transcription start, mRNA polymerization, binding of the ribosome to the mRNA, translation etc. (this list is certainly incomplete). Although we describe these as single steps, each of these events may consist of several molecular steps (e.g. mRNA polymerization alone consists of hundreds of individual steps) and may be rate-limiting or not.

As all steps starting with mRNA polymerization are shared by all promoters (since they all drive expression of the same fluorescent protein), it is the upstream steps that determine if a promoter has a high or low number of rate-limiting steps in our model. The estimated duration of a rate-limiting step also indicates that individual polymerization steps of mRNA production cannot be rate-limiting: A rate-limiting step for the most precise promoters is on the order of minutes (see page 5, paragraph 2 in the revised manuscript). This is far too long for the addition of single nucleotides in transcription or amino acids in translation, which take only tens of milliseconds.

If a promoter has a similar response time as the *LlacO-1* promoter, but a higher standard deviation (as the *cysK* promoter under NIT in our data set), it must have fewer rate-limiting steps with longer waiting times upstream of mRNA polymerization. To illustrate that increased average response times can coincide with fewer rate-limiting steps, we added a new experiment where the *LlacO-1* promoter is induced using IPTG at low concentrations. As a result, the response of this promoter is estimated to involve fewer rate-limiting steps, even though the sequence of steps activating this promoter is the same as for high IPTG concentrations (see Figure 2E). A plausible explanation for this phenomenon is that the low IPTG concentration makes steps such as IPTG uptake and binding to the lac repressor very slow. As a result, some steps that happened on a similar time scale at higher IPTG concentrations are now considerably faster (in relative terms) and thus have a smaller impact on overall timing variability.

In the revised manuscript, we explain this new experiment and its interpretation in the context of the queuing model on page 5, paragraph 3. We have also included a discussion of what such rate-limiting steps could be for our promoters and which molecular events can be ruled out as rate-limiting steps (page 5, paragraph 2 and page 8, paragraph 4).

6. One argument for the validity of the model derives from the observation that the response times follow an Erlang distribution (a special case of a gamma distribution). Many years ago it was already shown that a simple two-step model (transcription, translation) of stochastic gene expression leads to a gamma distribution (see, for example, Friedman et al., Phys. Rev. Lett. 97, 168302, 2006 or Shahrezaei & Swain, PNAS, 105, 17256, 2008) of protein concentrations. Furthermore, a more recent publication (Dal Co et al., NAR 45, 1069, 2017) shows that timing variability follows the steady-state protein distribution. You should comment on these alternative, rather simple models and put your proposal into perspective with this literature.

We now mention these papers together with a paper from Sunney Xie's lab that supports that the distribution of protein concentrations in *E. coli* is consistent with a gamma distribution (Taniguchi et al., 2010) in section "Response complexity and timing precision" of the Methods part (page 20, paragraph 2).

Minor points

7. All primary data (time series for all genes and cells) should be provided in SI.

We will include all primary time series data, as well as all quantitative data from the figures as supplementary material.

8. The microfluidics experiments involve the analysis of micro-colonies. Do you observe any effect on expression based on the location of the cell within the colony?

We could not detect any systematic dependence of single cell growth rate and cell morphology on the location within the microcolony. However, when correlating the distance from the edge of the microcolony with final expression level, we detected a slight correlation (Pearson correlation coefficient <0.5). This is a well-known and unavoidable technical issue in quantitative fluorescence imaging near the diffraction limit: the width of the point spread function is sufficient for cells to receive some fluorescence signal from their neighbors. As cells in the middle of the microcolony on average have more neighbors than cells on the edge of a microcolony, they receive more fluorescence signal from their neighboring cells, which sums up to a higher overall fluorescence. As we are capturing the response times of single cells at a relatively early stage of the microcolony (at 25% of the median full expression level) and this response time measure is independent of the maximal expression level of a single cell, this effect has a negligible impact on the response times we determine. We mention this effect now in the methods section (page 19, paragraph 1).

9. You normalize the expression threshold by the median full expression range of a colony. How does this expression range vary between colonies? This question is related to questions 1 above.

We have added a new supplementary figure (Figure EV6A), where we are depicting the range of the median full expression and the chosen threshold for each analyzed microcolony. For most promoters, there are only small differences between microcolonies.

2nd Editorial Decision

30 November 2018

Thank you again for sending us your revised work. We have now heard back from the three referees who agreed to evaluate your study. As you will see below, reviewers #1 and #2 are satisfied with the modifications made and think that the study is now suitable for publication. Reviewer #3 however, still raises strong concerns on the conclusions related to the number of rate-limiting steps. In particular, reviewer #3 thinks that substantial additional simulations would be required to support these theoretical predictions.

During our pre-decision cross-commenting process (in which the reviewers are given the chance to make additional comments, including on each other's reports), reviewers #1 and #2 felt that further clarifications and carefully toning down these theoretical claims would be sufficient to address the remaining issues listed by reviewer #3. The additional simulations suggested by reviewer #2 would

indeed significantly enhance the impact of the study. While we would not be opposed to the inclusion of such analyses should you feel inclined to perform them, we think that they are not mandatory for the acceptance of the study for publication. We would however ask you to perform text changes along the lines of the suggestions by reviewers #1 and #2.

REFEREE REPORTS

Reviewer #1:

The authors have addressed my concerns with this revision.

Reviewer #2:

The revised manuscript is much improved in clarity and rigor.

Reviewer #3:

Thank you for a detailed, point-by-point response to the remarks of all reviewers. Many technical points and analyses were clarified or extended. The revised text of the manuscript is better and I appreciate the new experiments and analyses. The overall conclusions remain unchanged.

Unfortunately, my overall appreciation also remains the same. I concur with the analysis of reviewer 2, stating that the work consists of "two parts that are not strongly integrated" (similar to my formulation of the two main conclusions drawn from the experiments). While I remain rather convinced of the analysis of the temporal ordering of reaction events (identification of a regulatory pathway), I am now even more doubtful about the conclusions concerning the number of rate-limiting steps.

Number of rate-limiting steps

In the previous version of the manuscript, the data were analyzed based on a model from queuing theory; which I found inappropriate. The model entails a strictly linear chain of events. Examples are enzyme reactions (a substrate can go through any number of intermediate states, but never becomes two substrates) or polymerization reactions (a DNA polymerase moves along the DNA, but a second polymerase will never be able to pass the polymerase in front). In my understanding, even the Aldous & Shepp reference is restricted in this sense. In such a restrictive context, I can conceive the theoretical predictions about the relationship between the randomness parameter and the number of rate-limiting steps.

In the present version of the manuscript "queuing theory" has been replaced by "statistical kinetics" and the reaction pathway "may be reversible, and the sequence can include branches and feedback loops". In other words, we are now dealing with a general Markov model (no memory transitions with exponential waiting time). I found it very surprising that timing variability (the randomness parameter) can be related analytically (at least as an inequality relationship) to the number of rate-limiting steps in such a general setting. Lacking an analytical tool, I performed some simulations analogous to the ones of Co et al. (2017). This simple model of gene expression contains two step: $\rightarrow \text{mRNA} \rightarrow \text{protein}$. Both mRNA and protein are degraded with respective rate constants. Using reasonable parameters for the rate constants (or transition probabilities in the Markov model), the kinetics resemble very much the kinetics measured by the authors and essentially reproduce the simulations by Co et al. (2017).

However, depending on the parameters chosen (especially the rate of production, I obtain a randomness parameter anywhere between 1 and 0.05. In other words, this simple, two-step mechanism can produce a randomness parameter inferior to what the authors claim requires 40 rate-limiting steps. This result can also be seen in the Co et al. publication. Therefore, unless I am missing some crucial information or still do not understand some fundamental concept, simple simulations (20 lines of code, plus 20 lines for nice graphics) disprove the claim that a small randomness parameter implies many rate-limiting steps. A crucial factor for low timing variability seems to be the promoter strength (number of proteins per cell in the induced state). Based on the simulations, I would predict a negative correlation between the randomness parameter and promoter strength. Since almost all curves are normalized, I could not verify this prediction. Non-normalized

data should be provided in SI. The newly added experiment of induction of the lac promoter at a lower IPTG concentration, as well as recA expression in TMP versus NIT are coherent with the predictions of the simple two-state model.

Ordering of reactions in a pathway and single-cell versus population experiments

I would also like to caution about definitive, i.e. general, conclusions concerning the ordering of reactions in a pathway by observing expression in single cells. I modified the initial model slightly to include an activator: $\rightarrow \text{mRNA1} \rightarrow \text{activator} \rightarrow \text{mRNA2} \rightarrow \text{protein}$. The signal leads to the production of a protein activator, which turns on the transcription of the target gene. All species have again appropriate degradation constants and the activator acts through a Michaelis-type function: the rate of production of mRNA2 is $k \cdot \text{activator} / (K_d + \text{activator})$. This model is clearly a sequential pathway. However, depending on the parameters, stochastic simulations show many cases where the protein crosses the threshold BEFORE the activator. This behavior is intuitively understandable. Once a certain number of molecules of activator are present in the cell, protein production can be very fast in particular cells. The stochasticity of the process necessarily produces cells in which the target protein crosses the threshold before the activator. Depending on the rate constants and the dissociation constant, K_d , I observe no overlap or frequent overlap. In other words: observing an overlap of the transition times of activator and target protein (Fig5B) does not conclusively rule out a sequential pathway. Conversely, given sufficiently separated time constants, parallel processes can lead to non-overlapping time profiles.

In summary

I am still convinced that the experiments are of very high quality. They are well carried out and interesting. However, I still question the conclusions drawn from the experiments. Simple models can explain the variability of the randomness parameter without invoking numerous rate-limiting steps. The dissection of the regulatory pathways are based on the observation of expression timing, but also on additional genetic evidence. Taken together, the conclusions concerning the signal transduction pathways are convincing. I would caution against very GENERAL conclusions solely based on expression timing.

Co AD, Lagomarsino MC, Caselle M & Osella M (2017) Stochastic timing in gene expression for simple regulatory strategies. *Nucleic Acids Res.* 45: 1069-1078

2nd Revision - authors' response

28 December 2018

Reviewer #3:

Thank you for a detailed, point-by-point response to the remarks of all reviewers. Many technical points and analyses were clarified or extended. The revised text of the manuscript is better and I appreciate the new experiments and analyses. The overall conclusions remain unchanged.

Unfortunately, my overall appreciation also remains the same. I concur with the analysis of reviewer 2, stating that the work consists of "two parts that are not strongly integrated" (similar to my formulation of the two main conclusions drawn from the experiments). While I remain rather convinced of the analysis of the temporal ordering of reaction events (identification of a regulatory pathway), I am now even more doubtful about the conclusions concerning the number of rate-limiting steps.

Number of rate-limiting steps

In the previous version of the manuscript, the data were analyzed based on a model from queuing theory; which I found inappropriate. The model entails a strictly linear chain of events. Examples are enzyme reactions (a substrate can go through any number of intermediate states, but never becomes two substrates) or polymerization reactions (a DNA polymerase moves along the DNA, but a second polymerase will never be able to pass the polymerase in front). In my understanding, even the Aldous & Shepp reference is restricted in this sense. In such a restrictive context, I can conceive the theoretical predictions about the relationship between the randomness parameter and the number of rate-limiting steps.

It is correct that most applications of statistical kinetics in the literature are different from the situation in our experiments: In gene expression, multiple mRNAs can be produced from one gene and multiple proteins from a single mRNA. The production of several of these molecules could correspond to multiple rate-limiting steps and we agree that this needs to be better explained. Still, we think that the statistical kinetics framework provides a useful context for our data. In particular, the estimated number of rate-limiting steps should be suitable for identifying relatively simple responses whose timing is controlled by few molecular steps. Importantly, our approach generates hypotheses and motivates new experiments aimed at identifying the key molecular steps controlling the timing variability of simpler responses.

In the revised manuscript, we now explicitly mention several times that the production of multiple mRNAs or proteins can correspond to multiple rate-limiting steps (page 4, paragraph 2,3; page 5, paragraph 3). We have further toned down or removed statements that the estimated number of effective rate-limiting steps may provide a quantitative measure of the complexity of the molecular chain of events leading to the observed gene expression changes (corresponding statements were removed or rephrased in the last sentence of the abstract; on page 4, paragraph 2 section heading; on page 4, paragraph 3; on page 6, paragraph 2; and on page 20, paragraph 2 section heading).

In the present version of the manuscript "queuing theory" has been replaced by "statistical kinetics" and the reaction pathway "may be reversible, and the sequence can include branches and feedback loops". In other words, we are now dealing with a general Markov model (no memory transitions with exponential waiting time). I found it very surprising that timing variability (the randomness parameter) can be related analytically (at least as an inequality relationship) to the number of rate-limiting steps in such a general setting.

Lacking an analytical tool, I performed some simulations analogous to the ones of Co et al. (2017). This simple model of gene expression contains two step: $\rightarrow \text{mRNA} \rightarrow \text{protein}$. Both mRNA and protein are degraded with respective rate constants. Using reasonable parameters for the rate constants (or transition probabilities in the Markov model), the kinetics resemble very much the kinetics measured by the authors and essentially reproduce the simulations by Co et al. (2017). However, depending on the parameters chosen (especially the rate of production, I obtain a randomness parameter anywhere between 1 and 0.05. In other words, this simple, two-step mechanism can produce a randomness parameter inferior to what the authors claim requires 40 rate-limiting steps. This result can also be seen in the Co et al. publication. Therefore, unless I am missing some crucial information or still do not understand some fundamental concept, simple simulations (20 lines of code, plus 20 lines for nice graphics) disprove the claim that a small randomness parameter implies many rate-limiting steps. A crucial factor for low timing variability seems to be the promoter strength (number of proteins per cell in the induced state). Based on the simulations, I would predict a negative correlation between the randomness parameter and promoter strength. Since almost all curves are normalized, I could not verify this prediction. Non-normalized data should be provided in SI. The newly added experiment of induction of the lac promoter at a lower IPTG concentration, as well as recA expression in TMP versus NIT are coherent with the predictions of the simple two-state model.

We thank the reviewer for proposing this alternative model for the interpretation of our data. It is important to note here that the so-called "two-step" model ($\rightarrow \text{mRNA} \rightarrow \text{protein}$) does not immediately correspond to two (or fewer) rate-limiting steps in the statistical kinetics framework: This model has many different states (specified by the number of mRNAs and proteins present, respectively) and accordingly many transition steps between these states. However, we agree that it is helpful to explain more clearly that the production of several mRNAs or proteins can correspond to multiple rate-limiting steps (see reply to previous point). In our experiments, the production of protein from mRNA is equivalent for every promoter studied, as the same ribosomal binding site and mRNA sequence (coding for a fluorescent protein) are used for every promoter. If promoter strength determined the differences in timing variability we observed, there should indeed be a negative correlation between the absolute expression level of a promoter and its timing variability, as a higher number of proteins reduces variability. We did not observe such a correlation in our data set.

To clarify this point, we have included a new figure panel (Figure EV2H). We also explain this point in the main text (page 5, paragraph 1). Here, we note that the absolute GFP expression level for the promoters is taken from our population-level measurements (Mitosch et al., 2017). As promoters had considerably different expression levels, we used different exposure times for stronger versus weaker promoters in the microscopy experiments. We now note this in the Methods section (page 18, paragraph 2), in the legend of Figure EV6, and in the supplementary data.

Ordering of reactions in a pathway and single-cell versus population experiments

I would also like to caution about definitive, i.e. general, conclusions concerning the ordering of reactions in a pathway by observing expression in single cells. I modified the initial model slightly to include an activator: $\rightarrow \text{mRNA1} \rightarrow \text{activator} \rightarrow \text{mRNA2} \rightarrow \text{protein}$. The signal leads to the production of a protein activator, which turns on the transcription of the target gene. All species have again appropriate degradation constants and the activator acts through a Michaelis-type function: the rate of production of mRNA2 is $k^* \text{activator}/(K_d + \text{activator})$. This model is clearly a sequential pathway. However, depending on the parameters, stochastic simulations show many cases where the protein crosses the threshold BEFORE the activator. This behavior is intuitively understandable. Once a certain number of molecules of activator are present in the cell, protein production can be very fast in particular cells. The stochasticity of the process necessarily produces cells in which the target protein crosses the threshold before the activator. Depending on the rate constants and the dissociation constant, K_d , I observe no overlap or frequent overlap. In other words: observing an overlap of the transition times of activator and target protein (Fig5B) does not conclusively rule out a sequential pathway. Conversely, given sufficiently separated time constants, parallel processes can lead to non-overlapping time profiles.

We thank the reviewer for this in-depth analysis performed in order to check whether unordered response times at the single-cell level always mean that the two respective steps are not in sequential order. We agree with the conclusions drawn by the reviewer. However, we want to reemphasize that, in addition to the temporal order, we quantify the correlation between the response times of the two promoters. We would not expect a strong positive correlation of the response times as observed for *ybjC* and *recA* under NIT (Figure 4C) for independent parallel processes with sufficiently different response times (page 8, paragraph 3). Further, even if the regulated promoter were detected before its activator in individual cells, a positive correlation between the response times of both promoters could still be observable. We mention in the Discussion that intrinsic noise may blur temporal order and response time correlations (page 8, paragraph 3).

In summary

I am still convinced that the experiments are of very high quality. They are well carried out and interesting. However, I still question the conclusions drawn from the experiments. Simple models can explain the variability of the randomness parameter without invoking numerous rate-limiting steps. The dissection of the regulatory pathways are based on the observation of expression timing, but also on additional genetic evidence. Taken together, the conclusions concerning the signal transduction pathways are convincing. I would caution against very GENERAL conclusions solely based on expression timing.

Co AD, Lagomarsino MC, Caselle M & Osella M (2017) Stochastic timing in gene expression for simple regulatory strategies. *Nucleic Acids Res.* 45: 1069-1078

We agree with the reviewer that some caution must be taken when applying the model from statistical kinetics to our data. We have reduced the prominence of the model in the revised manuscript and further toned down the interpretation of the data using the model throughout the text. Still, we think that the model from statistical kinetics provides an interesting context for our data and a basis for future studies and experiments.

Corresponding Author Name: Tobias Bollenbach

Manuscript Number: MSB-18-8470